# Rainfall and sea level drove the expansion of seasonally flooded habitats and associated bird populations across Amazonia

A. O. Sawakuchi [1] ✉, E. D. Schultz[2,10], F. N. Pupim[3], D. J. Bertassoli Jr.[4], D. F. Souza[5], D. F. Cunha[6], C. E. Mazoca[1], M. P. Ferreira [6], C. H. Grohmann [7], I. D. Wahnfried [8], C. M. Chiessi [4], F. W. Cruz[1], R. P. Almeida[1] & C. C. Ribas[2,9]

Spatial arrangement of distinct Amazonian environments through time and its effect on specialized biota remain poorly known, fueling long-lasting debates about drivers of biotic diversification. We address the late Quaternary sediment deposition that assembled the world's largest seasonally flooded ecosystems. Genome sequencing was used to reconstruct the demographic history of bird species specialized in either early successional vegetation or mature floodplain forests. Sediment deposition that built seasonally flooded habitats accelerated throughout the Holocene (last 11,700 years) under sea level highstand and intensification of the South American Monsoon, at the same time as global increases in atmospheric methane concentration. Bird populations adapted to seasonally flooded habitats expanded due to enlargement of Amazonian river floodplains and archipelagos. Our findings suggest that the diversification of the biota specialized in seasonally flooded habitats is coupled to sedimentary budget changes of large rivers, which rely on combined effects of sea level and rainfall variations.

Amazonia hosts the largest biodiversity on Earth[1], probably due to its high environmental heterogeneity and dynamic landscapes[2,3]. The relationship between landscape evolution and Neotropical species diversification has been intensely debated during the last decades[2]. Time-calibrated phylogenetic data show that most Amazonian extant species emerged during the late Pliocene and Pleistocene (last ~5 Ma)[2–5]. However, the mechanisms through which populations respond to changes in the physical environment are poorly constrained in time and space. Previous studies that interpret biotic diversification driven by landscape changes consider geological or climatic processes that operate in periods of million to thousand years such as mountain building since the Miocene[6], development of large rivers during the Pliocene and Pleistocene[7], or climate-forced shifts in upland forests distributions during the Pleistocene[8]. Often, lack of congruent spatial and, especially, temporal phylogenetic patterns have led to the alternative interpretation that landscape change had a minor role in Amazonian diversification, with most diversification driven by long-distance dispersal or intrinsic biotic processes[9]. However, both

[1]Institute of Geosciences, University of São Paulo, Rua do Lago 562, São Paulo, SP, Brazil. [2]Programa de Pós-Graduação em Biologia (Ecologia), Instituto Nacional de Pesquisas da Amazônia, Av. André Araújo 2936, Manaus, AM, Brazil. [3]Departamento de Ciências Ambientais, Universidade Federal de São Paulo (UNIFESP), Rua São Nicolau 210, Diadema, SP, Brazil. [4]School of Arts, Sciences and Humanities, University of São Paulo, Av. Arlindo Bettio 1000, São Paulo, SP, Brazil. [5]Gerência de Hidrologia e Gestão Territorial, Serviço Geológico do Brasil (CPRM-SGB), Rua Costa 55, São Paulo, SP, Brazil. [6]Programa de Pós-Graduação em Geoquímica e Geotectônica, Institute of Geosciences, University of São Paulo, Rua do Lago 562, São Paulo, SP, Brazil. [7]Institute of Energy and Environment, University of São Paulo, Av. Prof. Luciano Gualberto 1289, São Paulo, SP, Brazil. [8]Departamento de Geociências, Universidade Federal do Amazonas, Av. Gen. Rodrigo Octávio Jordão Ramos 6200, Manaus, AM, Brazil. [9]Coordenação de Biodiversidade, Instituto Nacional de Pesquisas da Amazônia, Av. André Araújo 2936, Manaus, AM, Brazil. [10]Present address: Department of Ornithology, American Museum of Natural History, 200 Central Park West, New York, NY, USA. ✉e-mail: andreos@usp.br

views are limited by the scarcity of chronological information (i) to constrain the presumed environmental changes that split populations or (ii) to support landscape stability during the diversification of specific taxa. Thus, the main challenge to test whether environmental changes drove biotic diversification is reconstructing the landscape and biotic histories under the same spatiotemporal framework.

Amazonia is a mosaic of landscapes and ecosystems[10,11], where the South American Monsoon System (SAMS) and the continental-scale river system determine the distribution of two end-member types of environments:[10,12] non-flooded uplands (*terra firme*) and seasonally flooded habitats (*várzea and igapó*). Sediments accumulated within valleys of large Amazonian rivers shape substrates that are colonized by vegetation adapted to seasonal flooding[12], originating the world's largest flooded ecosystem that covers an area of $8.4 \times 10^5$ km$^2$ [13]. The Amazonian seasonally flooded habitats sustain a rich and specialized biota[13]. Among the 4962 Amazonian tree species, 2166 species occur in seasonally flooded habitats, and up to 30% may be endemic to these habitats[14–16]. Besides, more than 150 species of Amazonian nonaquatic birds are restricted to or highly dependent on seasonally flooded habitats, including several species specialized in river islands[17–20]. The distribution and diversification of organisms adapted to seasonally flooded ecosystems are possibly linked to the current and past availability and connectivity of floodplains and islands built by sediment deposition within major river valleys[21]. The dynamics of sediment transport and accumulation that create or erode alluvial plains drive the expansion (or fragmentation) of Amazonian flooded substrates in a thousand-year timescale[22]. Hence, the spatial distribution and population dynamics of species adapted to seasonally flooded habitats are

presumably dependent on the sedimentary budget of Amazonian rivers, but data necessary to test the synchronicity between historical environmental changes and the biotic response are absent so far.

Here, we determine the formation chronology of seasonally flooded habitats in lowland Amazonia (Figs. 1 and 2) using sediment deposition ages obtained by optically stimulated luminescence (OSL) and radiocarbon dating methods. We obtained OSL ($n = 93$) and radiocarbon ($n = 23$) ages that were combined with ages ($n = 53$) compiled from the literature (Supplementary Data 1) to constrain the chronology of sediment accumulation in extensive areas occupied by seasonally flooded habitats in the whitewater Solimões and Amazon Rivers (*várzea*) as well as in major blackwater and clearwater tributaries (*igapó*). This includes the formation chronology of the Anavilhanas archipelago in the Negro River, and the Tabuleiro do Embaubal archipelago in the Xingu River, respectively the largest archipelagos in blackwater and clearwater tributaries of the Amazon River.

In parallel, we reconstruct the demographic history of populations of birds specialized in seasonally flooded habitats in early (young islands and river banks) and late (mature floodplain forest) vegetation successional stages using reduced genome sequencing (ultraconserved elements, UCEs)[23]. Birds are the vertebrate group for which species limits and distributions are best known, and thus this group has frequently been used to infer biogeographic processes in Amazonia, where biodiversity patterns for many other animal groups are still poorly understood[2]. We reconstruct past demography for populations of five bird species specialized in early successional habitats, mainly in river islands, and four species specialized in floodplain forest environments (Supplementary Fig. 1). This

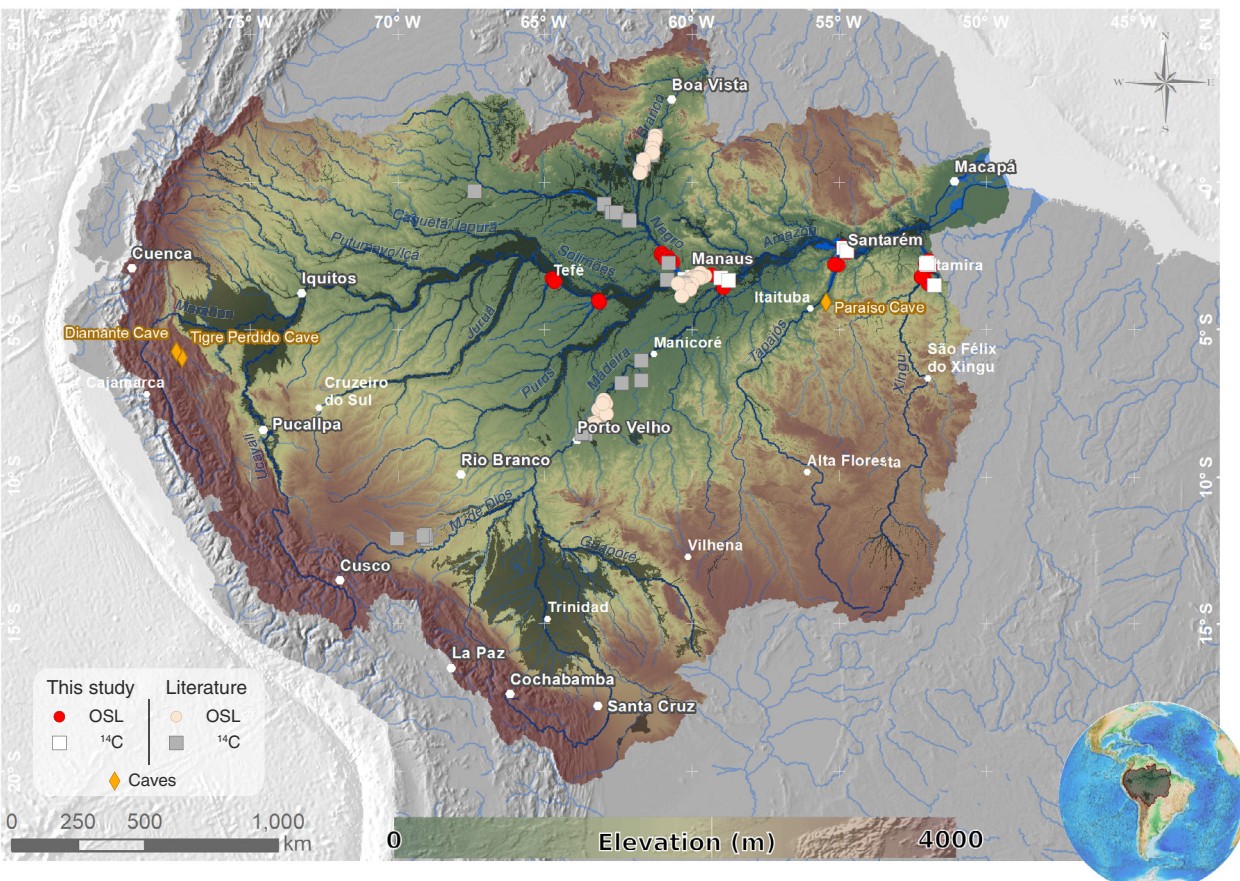

**Fig. 1 | Sites with OSL and radiocarbon (¹⁴C) ages from substrates of seasonally flooded areas (dark green shade) across the Amazon River basin.** Sites sampled in this study can be observed in more detail in Fig. 2. Geographical coordinates of samples and ages data are shown in Supplementary Data 1. The Tigre Perdido, Diamante and Paraíso caves correspond to sites with paleoprecipitation reconstructions presented in Figs. 7 and 8.

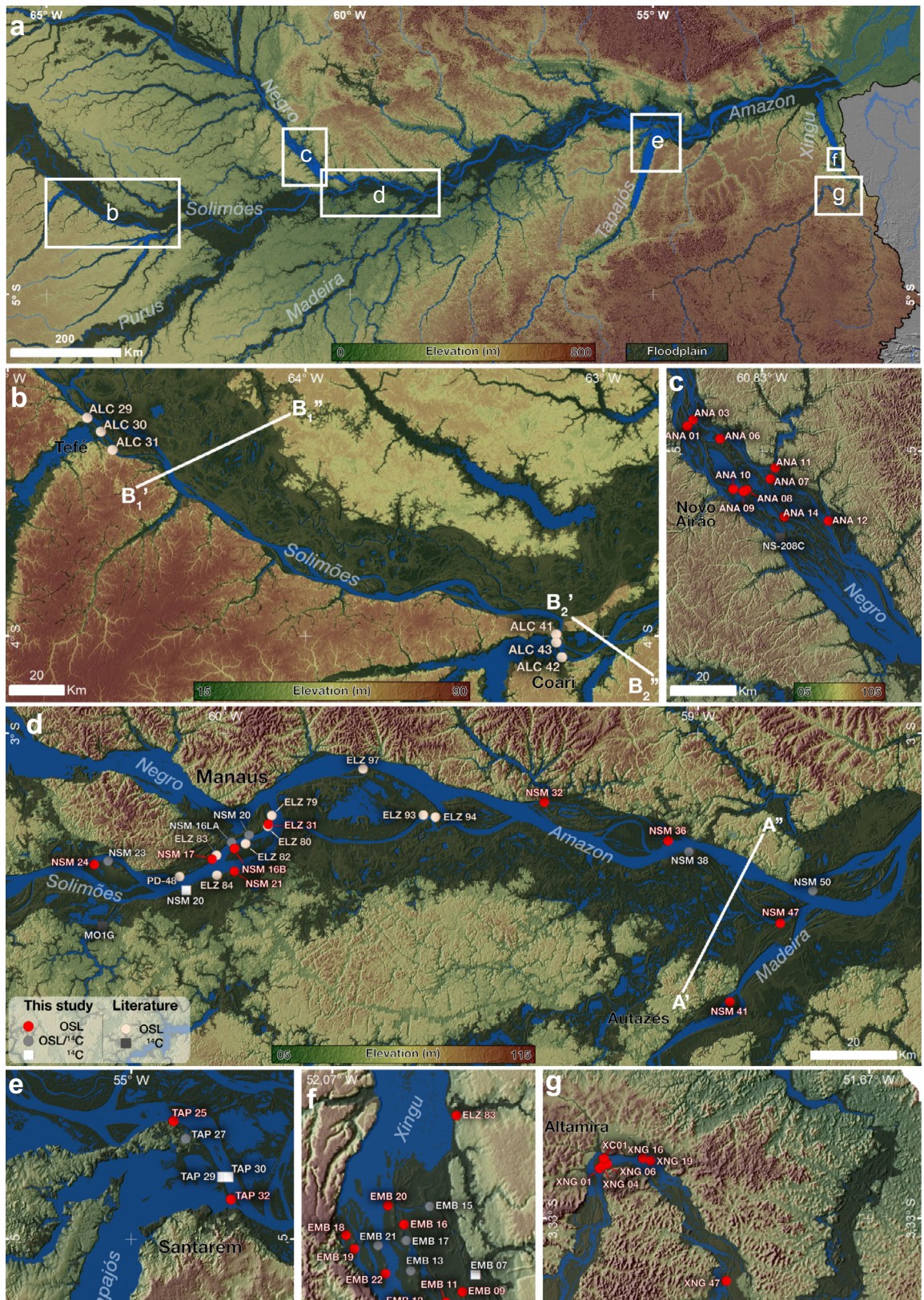

**Fig. 2 | Seasonally flooded areas (in green shades) within the Solimões-Amazon River valley and main tributaries in western, central and eastern Amazonia.** Upland areas are shown in yellow-red shades. **a** White rectangles highlight areas sampled for OSL and radiocarbon ($^{14}$C) dating in this study. Elevation profiles A′–A″, B₁′–B₁″ and B₂′–B₂″ are shown in Fig. 6. **b** Solimões River main stem between the Tefé and Coari Rivers. **c** Anavilhanas archipelago in the Negro River. **d** Solimões-Amazon River main stem and lower Madeira River. **e** Downstream sector of the Tapajós River and confluence with the Amazon River. **f** Tabuleiro do Embaubal archipelago in the lower Xingu River. **g** Xingu River in the Brazilian Shield. Main cities are shown for location.

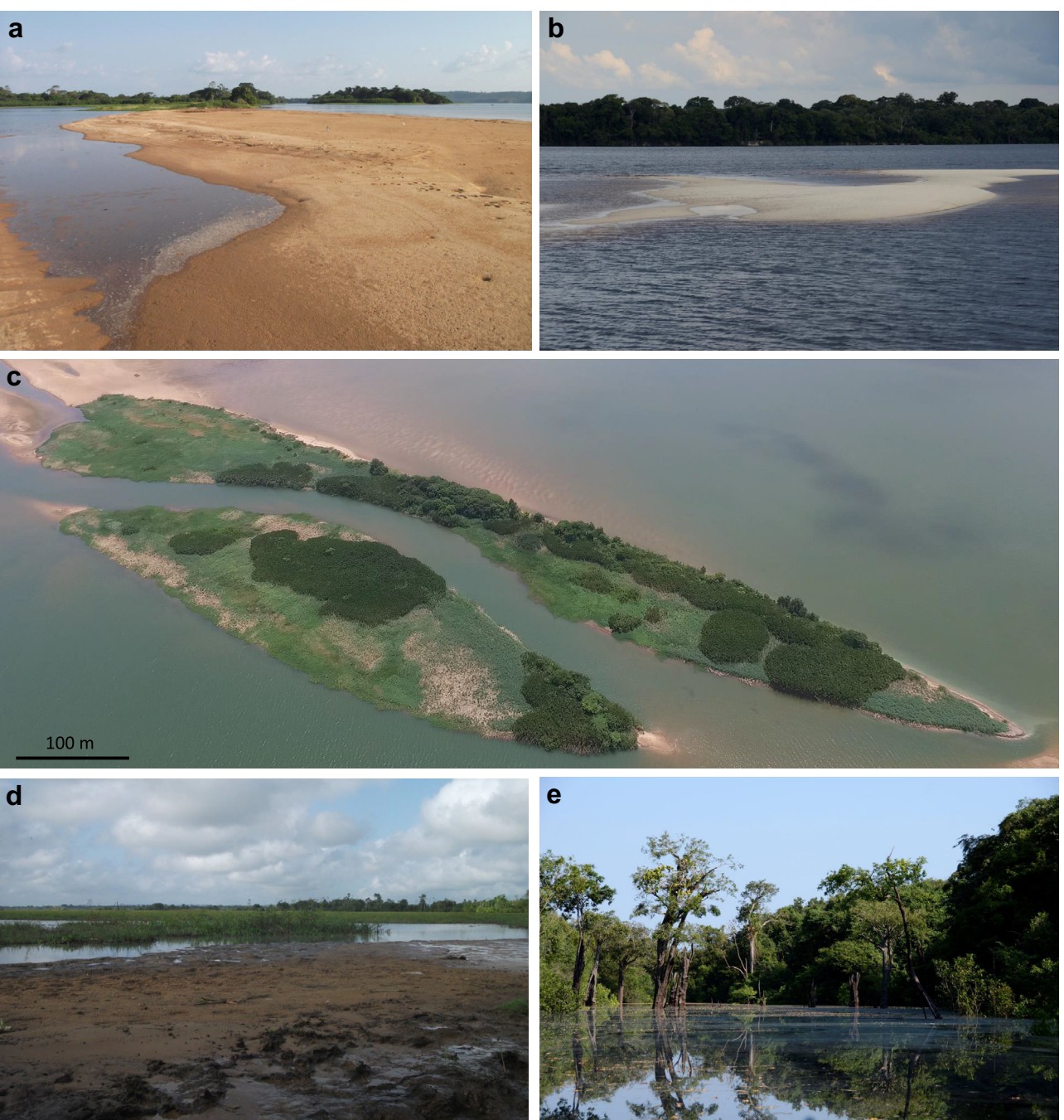

**Fig. 3 | Development of seasonally flooded vegetation over sandy bar. a**, **b** Sand bar top exposed during the low-stage water level. **c** Sand bar partially colonized by grass and bush vegetation. **d** Sand bar top with deposition of silt and clay on slack water zone formed by vegetation growth. **e** Mature flooded forest (*igapó*) developed over stabilized sand bars. **a**, **c**, **d** correspond to sites in the Tabuleiro do Embaubal archipelago, Xingu River. **b**, **e** are sites in the Anavilhanas archipelago, Negro River.

approach allows testing for chronologically congruent responses between bird populations and their habitats. Thus, this study characterizes the response of animal populations associated with different Amazonian seasonally flooded habitats to long-term changes in the physical landscape. The notion of how the Amazonian biota responds to environmental changes is crucial to understanding evolutionary processes and determining conservation policies under future climate change scenarios and the intensification of regional anthropogenic threats. Additionally, the Amazonian floodplains represent a major natural source of methane ($CH_4$) to the atmosphere[24,25]. Hence, tracking periods of expansion or retraction of the Amazonian floodplains allows the appraisal of potential connections between the sedimentary budget of large tropical rivers and global climate.

## Results

### Sedimentology of seasonally flooded substrates

Seasonally flooded habitats within large Amazonian river valleys expand when sand bars are exposed and colonized by vegetation during periods of low-stage water level (Fig. 3a–c). Pioneer vegetation increases the trapping of fine-grained suspended sediments that originates cohesive surfaces, enabling the development of successional

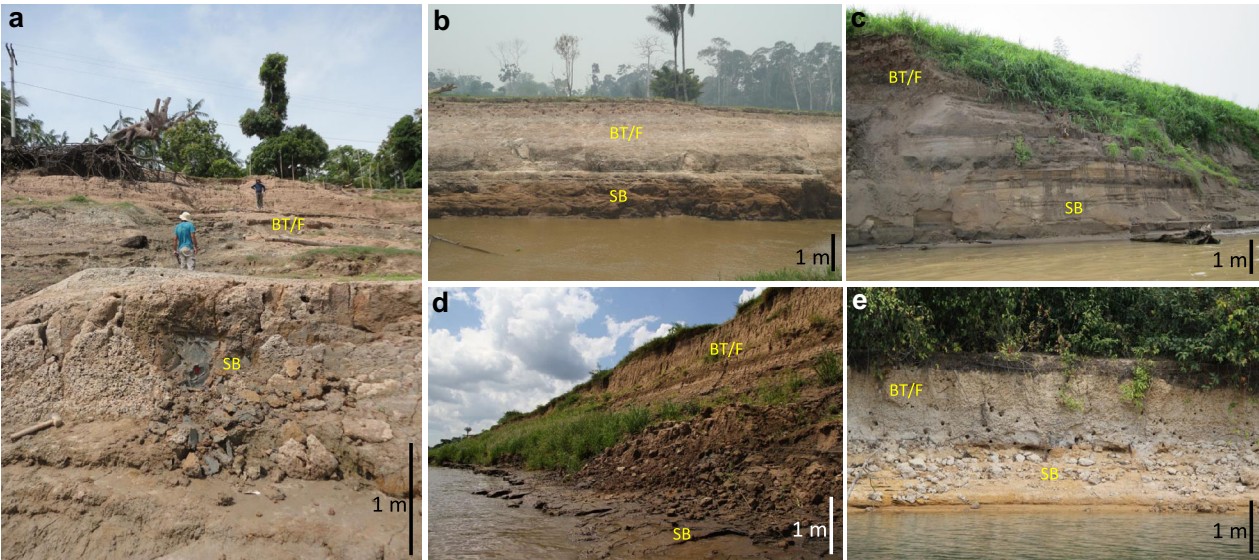

**Fig. 4 | Sedimentary deposits forming the substrates of seasonally flooded environments in lowland Amazonia.** The general sedimentary architecture is represented by underlying sandy bar (SD) deposits covered by muddy bar top and floodplain (BT/F) deposits. Pictures were taken during the dry season (September to November). Sedimentary facies are shown in Fig. 6. **a** Right margin of the Solimões River, upstream the Tefé River mouth near sampling site ALC29. **b** Left margin of the Solimões River, near the Purus River mouth. **c** Right margin of the Solimões River, upstream the Negro River mouth near sampling site NSM20. **d** Right margin of Amazon River upstream of the Tapajós River mouth, near sampling site TAP27. **e** Island of the Xingu River, near the city of Altamira. Scale is approximate. Geographic coordinates of sampling sites are presented in Supplementary Data 1.

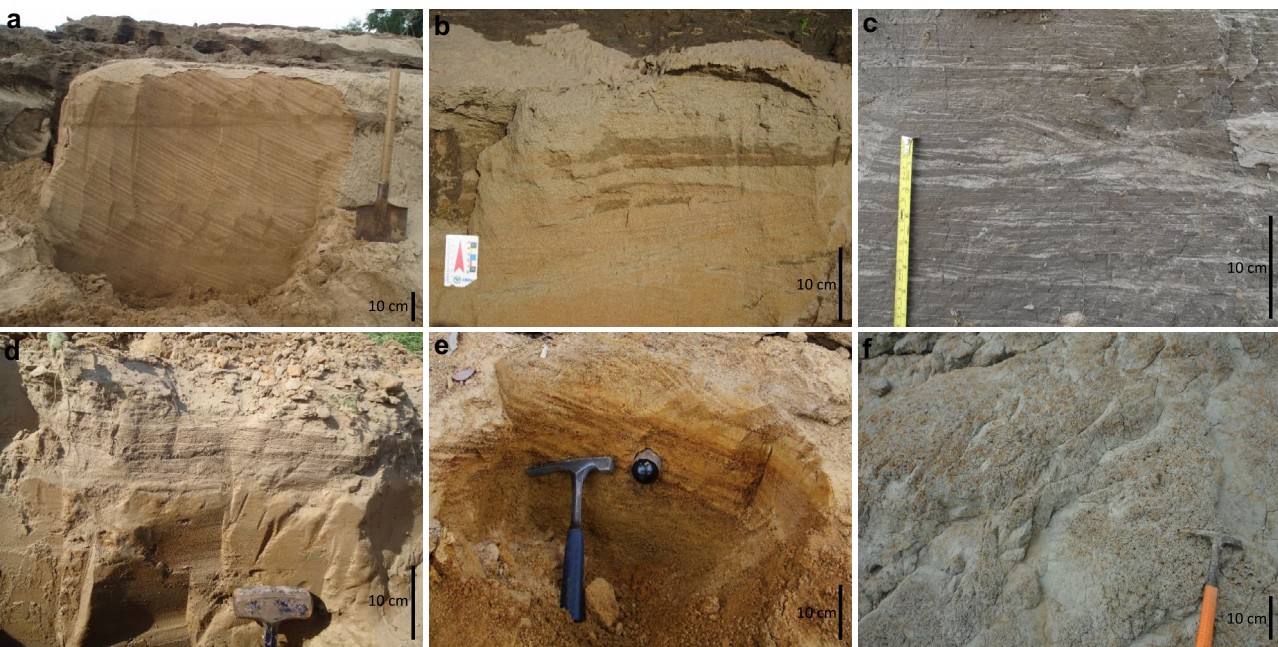

**Fig. 5 | Major sedimentary facies representing deposits of the seasonally flooded environments.** **a** Sets of fine sand with tabular cross stratification (Amazon River near the Madeira River mouth). **b** Fine sand with tabular cross stratification underlying fine sand layer with low-angle stratification and cross lamination (Solimões River near the Negro River mouth). **c** Sandy mud with ripple and horizontal heterolithic lamination (Amazon River near the Tapajós River mouth). **d** Fine sand with tabular cross stratification underlying a set of low-angle stratification (downstream reach of the Tapajós River near Santarém). **e** Coarse sand with tabular cross stratification (Xingu River near Altamira). **f** Massive mud (Solimões River near Coari).

forests within years to decades, that eventually form mature seasonally flooded forests (Fig. 3d, e). The sedimentary substrates of seasonally flooded habitats related to Amazonian rivers generally encompass sand layers covered by or intercalated with mud layers (Figs. 4 and 5). These substrates comprise sedimentary deposits formed and exposed between the low-stage and high-stage water levels, thus representing upper bar deposits[26]. In general terms, the sedimentary successions display marked fining-upward trends and are coarser-grained at the bar-heads. The typical succession representing the progressive conversion of migrating sand bars into stabilized marginal floodplains or islands covered by seasonally flooded vegetation can be described as follows:

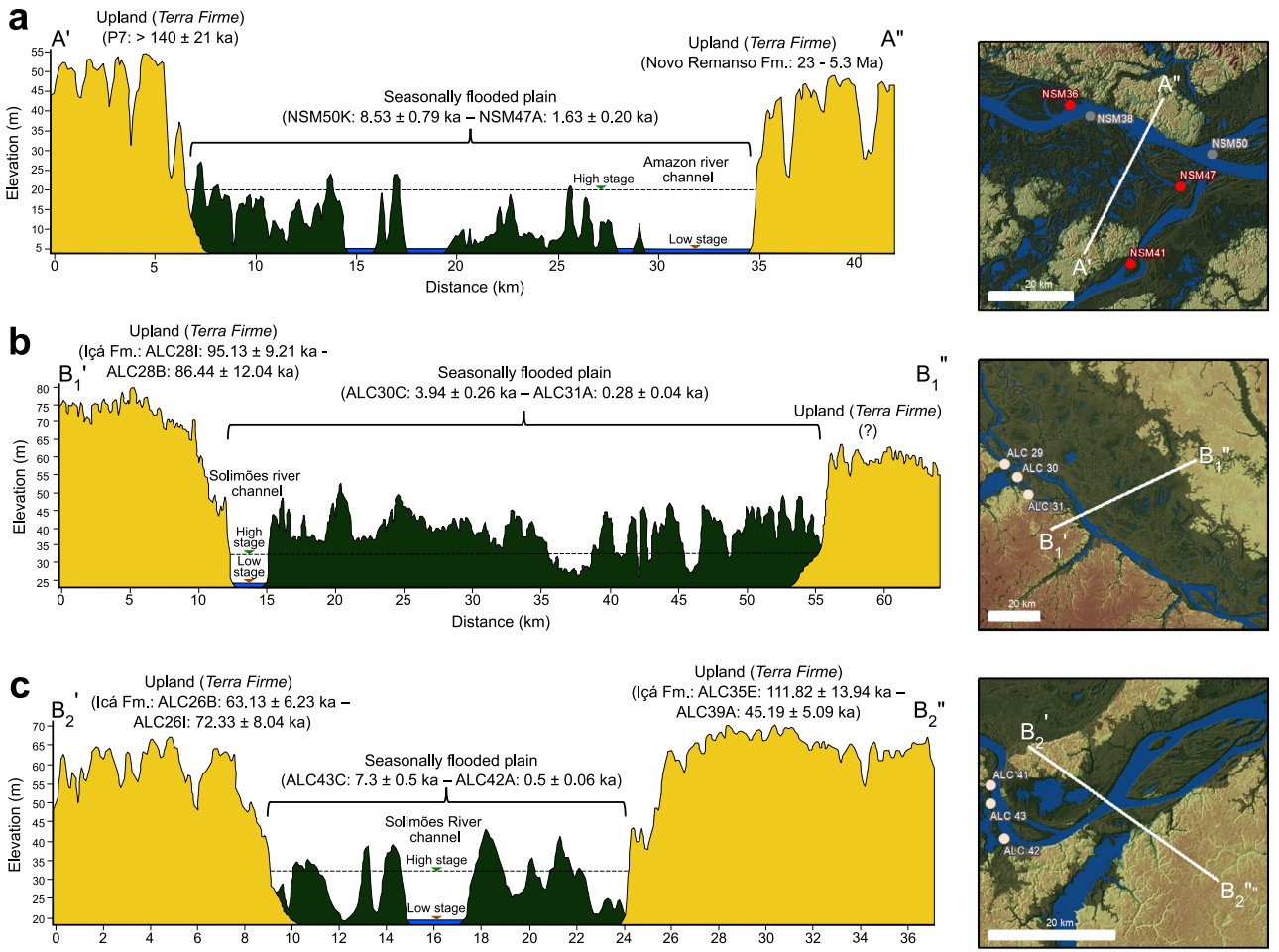

**Fig. 6 | Elevation profiles showing the morphostratigraphic relation between upland fluvial terraces and seasonally flooded plains within the Solimões-Amazon River valley. a** Amazon River main stem near the Madeira River mouth (see location in Fig. 2a, d). Fluvial deposits forming upland terraces correspond to the Novo Remanso Formation (Fm.) and undifferentiated Pleistocene sediments[63]. **b**, **c** Solimões River main stem near Tefé and Coari (see location in Fig. 2a, b). Ages attributed to the upland terraces correspond to the Içá Formation (Fm.) in the study

area[22]. In all profiles, the age ranges are based on OSL and radiocarbon ages obtained in this study. Approximate elevations of high and low water stages were determined through historical values recorded by gauge stations (Agência Nacional de Águas - HidroWeb - Sistema de Informações Hidrológicas: http://snirh.gov.br/hidroweb/[accessed 20 January 2022]) nearby the areas. High frequency variations in the elevation profiles record the vegetation covering the upland and seasonally flooded areas.

**Subaqueous dune deposits (Sand Bar facies, SB).** Tabular to trough cross-stratified sand disposed in co-sets with inclined cross-set bounding surfaces (Fig. 5a), mostly dipping downstream, but also laterally and even upstream to the flow direction[26]. In bar-head areas, coarse sand beds mark the cross-strata foresets (Fig. 5e), whilst fine sand dominates bar tail areas (Fig. 5a, b). These deposits are found near the low-stage water level. They are formed by tens of meters long straight-crested and sinuous-crested dunes visible in high-resolution bathymetric surveys[27,28] and exposed during the low-stage water level in bar margins. Local reworking by shallow currents or winds results in low-angle cross-strata in fine sand (Fig. 5b). These sandy sedimentary successions are typically a few to several meters thick.

**Current ripple deposits (Bar Top facies, BT).** Muddy fine sand and silt with abundant, mostly stoss-preservational, cross lamination (Fig. 5c) arranged in tabular beds ranging from a few decimeters to a few meters in thickness, mainly in bar tails and downstream areas of bars. These deposits are formed by the downstream migration and vertical aggradation of fine-grained sediments in areas of flow deceleration, in the presence of ripple-scale bedforms, and overlaying cross-stratified sandy deposits.

**Mud settling and trapping (Floodplain facies, F).** The uppermost deposits of the investigated sedimentary successions, where preserved, are characterized by structureless mud with varied organic matter content (Fig. 5f). These deposits are mostly formed by fine-grained particles trapped by bar-top vegetation during high-stage water level and can reach up to several meters thick.

## Chronology of sediment deposition

The stabilization of fluvial sedimentary substrates records the occupation of the newly formed river islands or marginal floodplains by vegetation. Thus, the timing of vegetation establishment can be accessed by OSL and radiocarbon ages obtained in SB or BT facies that occur in substrates of seasonally flooded habitats. The belt of seasonally flooded habitats along the Solimões River is bounded by elevated late Pleistocene fluvial terraces, with sediment deposition between 120 and 45 ka, covered by upland *terra firme* forests[22]. Floodplains occupied larger and higher elevation areas (Fig. 6) during most of the late Pleistocene and probably experienced a pronounced retreat after 45 ka due to regional channel incision[22]. This pattern agrees with our geochronological dataset that suggests the inception of a new phase of floodplain expansion since the Last Glacial Maximum

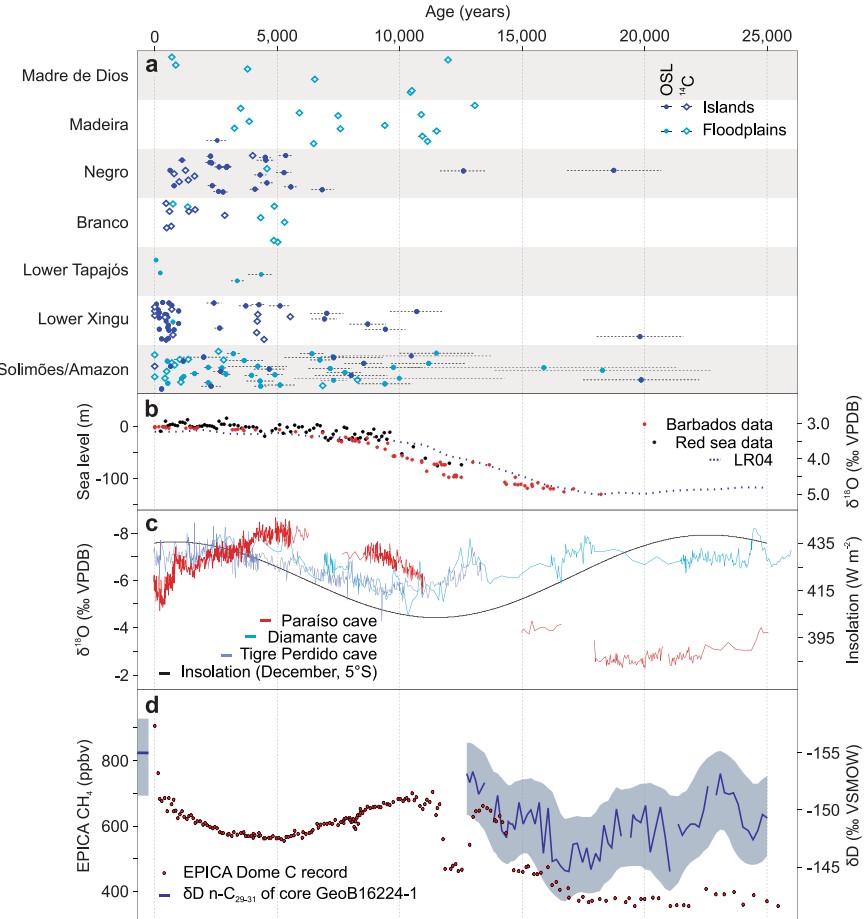

**Fig. 7 | Summary of results (last 25 ka).** OSL and radiocarbon ages (**a**) are compared to sea level variation (**b**), precipitation records across Amazonia (**c**, **d**) and atmospheric methane ($CH_4$) concentration (**d**). OSL and radiocarbon ages have an offset in the y-axis to avoid symbol overlap and allow visualization. In **a**, dots represent sediment sample ages and horizontal bars represent age errors calculated according to the Gaussian law of error propagation. Sea level variation combines global records[40–42] and is considered representative of the equatorial Atlantic Ocean. Precipitation records (**c**) correspond to stable oxygen isotopes ($\delta^{18}O$) measured in speleothems from western[44] and eastern[53] Amazonia and stable hydrogen isotopes measured in plant wax ($\delta D_{n-C29−31}$) (**d**) from a marine sediment core (GeoB16224-1) offshore the outflow of the Amazon River[52]. Atmospheric concentrations of $CH_4$ (**d**) correspond to data from Antarctic ice cores[64,65]. Precipitation records across Amazonia correspond to speleothems from the Diamante, Tigre Perdido and Paraíso caves, whose locations are shown in Fig. 1. OSL and radiocarbon ages generated in this study and compiled from published works (**a**) are presented in the Supplementary Information.

(LGM) around 21 ka, but with acceleration during the Holocene (last 11.7 ka) (Figs. 6, 7a, 8a, Supplementary Data 1 and 2). The regional character of this trend is also supported by sediment deposition ages obtained in floodplains of the Madre de Dios River in westernmost Amazonia[29], Branco River in northern Amazonia[30], and downstream reaches of the Madeira and Solimões Rivers in central Amazonia[31,32] (Fig. 8).

Sediment accumulation rates decrease from $10^0$ cm yr$^{-1}$ in the decadal timespan to $10^{-2}$ cm yr$^{-1}$ in the millennial timespan (Supplementary Figs. 2, 3, Supplementary Data 3), pointing to valley filling characterized by short term pulses of sediment accumulation, whose deposits are further reworked by recurrent erosion events. This sedimentation pattern is evident in the development of the Anavilhanas (Supplementary Figs. 4 and 5) and Tabuleiro do Embaubal (Supplementary Figs. 6, 7, and 8) archipelagos respectively in the incised valleys of the Negro and Xingu Rivers. Both archipelagos represent valley filling during the Holocene, but specific islands were formed in shorter time periods around 4 ka, 2.5 ka, and from 1 ka to present (Supplementary Figs. 4 and 6). Sedimentary islands younger than 1 ka have mature arboreal vegetation, indicating that the late-successional flooded forest stage can be attained in a centennial timespan. On the local scale ($10^2$–$10^3$ m), fluvial deposits representing substrates

colonized by seasonally flooded vegetation in the Solimões and Amazon Rivers also show depositional and colonization rates varying from centennial to millennial timescales.

## Demography of bird populations

The recent population dynamics of birds specialized in two kinds of seasonally flooded habitats, represented by early successional vegetation habitats in river islands and mature floodplain forests, is temporally congruent with the Holocene expansion of floodplains and archipelagos after the LGM (~21 ka) as indicated by our geochronological dataset (Fig. 8). Genomic data (average of 2233 UCE loci and 1574 SNPs and two mtDNA loci per taxon) were obtained for 210 individuals belonging to nine bird species, including five specialized in early successional vegetation, which occur mainly in river islands (*Cranioleuca vulpecula*, *Furnarius minor*, *Knipolegus orenocensis*, *Mazaria propinqua*, and *Stigmatura napensis*) and four species specialized in floodplain forests (*Attila bolivianus*, *Cranioleuca gutturata*, *Myiopagis flavivertex*, and *Nasica longirostris*)[17,33]. Isolation by distance (IBD) and population structure generating unequal gene flow among individuals are known to potentially mislead demographic inferences[34]. We show for all taxa studied here that regions of lower and higher migration differ from IBD expectations (Supplementary

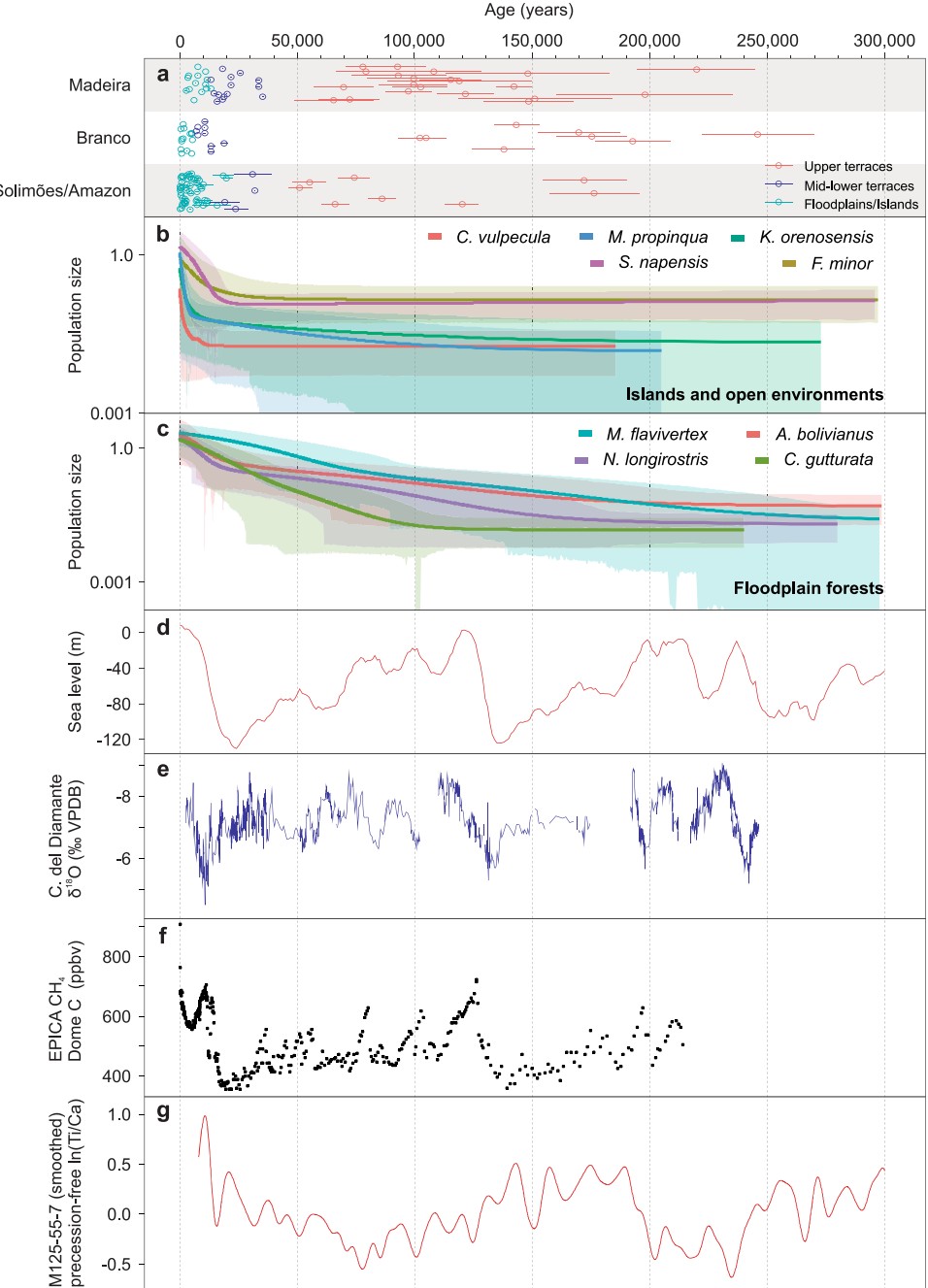

**Fig. 8 | Summary of results (last 300 ka).** OSL and radiocarbon ages from flood-plains/islands and fluvial terraces (**a**) are compared to plots showing population sizes through time estimated in the EBSP analyses for bird species specialized in islands or riverbanks (**b**), and for bird species specialized in floodplain forests (**c**), sea level variation derived from benthic foraminifera δ18O stack from globally distributed marine sites (**d**), precipitation variation in western Amazonia (**e**), atmosphere methane (CH4) concentration (**f**) and precession-free variation of titanium-to-calcium ratio (ln(Ti/Ca)) in marine sediments (core M125-55-7) offshore eastern Brazil (**g**). Data generated in this study (**a**, **b**, **c**) are presented in Supplementary Information. In panel **a**, dots represent sediment sample ages and horizontal bars represent age errors calculated according to the Gaussian law of error propagation.

The unit named as "Mid-Lower terraces" in panel **a** can include higher elevation still susceptible to flooding. Confidence intervals (shaded areas) in **b**, **c** correspond to 95% High Posterior Density. Higher amplitude variations (**d**) in stable oxygen isotopes (δ18O) stack from benthic foraminifera[42] indicate sea level changes representative of the equatorial Atlantic Ocean. Precipitation variation in western Amazonia (**e**) corresponds to stable oxygen isotopes (δ18O) measured in spe-leothems from the Diamante cave[44]. Atmospheric concentrations of CH4 (**f**) correspond to data from Antarctic ice cores[64,65]. Precession-free variation of ln(Ti/Ca) in marine sediments offshore eastern Brazil (**g**) represents the intensity of the SAMS[58].

Fig. 9) when Estimated Effective Migration Surface (EEMS) is applied. In addition, two tests for the best fit number of genetic clusters within each species, performed using sparse Non-Negative Matrix Factorization (sNMF) and k-means, consistently indicate absence of intra-specific population structure. The only exception is *K. orneocensis*, which shows two genetically distinct populations corresponding the

described subspecies *K. o. sclateri* and *K. o. xinguensis* (Supplementary Figs. 1, 9 and 12).

Two different demographic reconstruction methods consistently suggest demographic expansion for all studied species. In the Extended Bayesian Skyline Plots (EBSP), population stability is rejected, as the 95% confidence interval of the population size change parameter

does not include 0 for all studied taxa, with the exception of *F. minor*. Furthermore, the statistical model comparisons performed using Demographic Inferences with Linked Selection (DILS[35]) recovered models of population expansion instead of retraction or stability, with high posterior probabilities for all taxa (>0.98) (Supplementary Data 4). Moreover, we show that there is a distinct temporal pattern of demographic expansion in the two groups of species with distinct habitat affinities. Mature floodplain forest specialists had independent and continued expansion since around 350–100 ka, which was enhanced by a second expansion after the LGM (~21 ka). The specialists in early successional habitats experienced a recent shared expansion in the last 21 ka (Fig. 8b and Supplementary Fig. 10). This model is supported by the two aforementioned inferences based on individual taxa and by an additional Bayesian test of co-expansion in *Ecoevolity*[36]. *Ecoevolity* results suggest six distinct events of demographic expansion, being four of them related to independent and older demographic changes in floodplains forests taxa, and two more recent events of demographic changes on island-specialists, one shared by four taxa (*F. minor*, *K. o. sclateri*, *M. propinqua*, and *S. napensis*), which was followed by a slightly more recent expansion on *C. vulpecula* (Supplementary Figs. 10, 11 and 13). The onset of demographic expansion for the island-specialists based solely on the UCEs dataset (DILS and *Ecoevolity*) dates to 30–20 ka ago (Supplementary Figs. 10 and 11), while the demographic reconstructions calibrated by both UCE rates and the better-known rates for mtDNA markers indicate a slightly younger age for these demographic expansions, after the LGM, and increasing into the Holocene (Fig. 8b).

## Discussion

### Expansion and retraction of seasonally flooded habitats

Sediment deposition ages suggest that the Holocene expansion of seasonally flooded habitats, represented by marginal floodplains and islands with early successional vegetation to mature forests, was ubiquitous and synchronous throughout lowland Amazonia. The extension of seasonally flooded areas within lowland river valleys through time relies on variations of erosion base level and sediment supply[37]. Although autogenic fluvial processes lead to heterogeneous filling of river valleys in decadal to secular timescales, as pointed out by the formation chronology of islands in the Xingu and Negro Rivers (Supplementary Figs. 4–8), the regional expansion of seasonally flooded substrates across lowland Amazonia was driven by synergistic effects of erosion base level and fluvial discharge on sediment accumulation in millennial-scale timespans. Changes in sea level control the erosion base level in coastal low-elevation fluvial systems, but its effects are attenuated over upstream areas[37,38]. Global sea-level rise started abruptly after the LGM[39], favoring sediment accumulation in major lowland river basins, especially when sea level reached a highstand condition during the Holocene. Our geochronological dataset indicates expansion of Amazonian seasonally flooded habitats during the Holocene (Figs. 6, 7a and 8a), coevally with a sea-level highstand phase[40–43] (Fig. 7b) favorable to sediment aggradation in the downstream sector of the Amazon River. However, the floodplains and archipelagos growth during the Holocene across lowland Amazonia (Fig. 7a) occurred in areas thousands of kilometers away from the coastline (Figs. 1, 2 and 6), where the effect of sea level on sediment deposition or erosion is expected to be reduced and retarded. Additionally, the Holocene was also marked by the orbitally-driven gradual strengthening of the SAMS[44,45] (Figs. 7c, 8e and 8g), that probably increased water discharge and riverbank erosion, competing with the eventual sediment trapping forced by the sea-level highstand condition.

Phases of high sediment input could also promote the expansion of seasonally flooded substrates through temporary storage of sediments in river valleys. The sediment discharge of the Solimões-Amazon River is mainly derived from the Andes[46,47], with interpreted

millennial-scale variations coupled to precipitation changes of the SAMS[48,49]. Western Amazonia, including the Andean headwaters of the Solimões-Amazon River, experienced a higher monsoonal rainfall both during the LGM and middle to late Holocene[48,50]. On the other hand, climate records from northern and eastern lowland Amazonia, which comprise sediment sources of the Negro and Xingu Rivers, show an opposite response, with a drier LGM and wetter early to middle Holocene[51–55] (Fig. 7c, d). As in the case of sea-level forcing, the heterogeneous rainfall history alone cannot explain the synchronous expansion of seasonally flooded habitats in rivers draining western, northern, central, and eastern lowland Amazonia.

In summary, the synchronous expansion of seasonally flooded habitats across lowland Amazonia during the Holocene cannot be attributed exclusively to sea level or rainfall variations. Islands with mature seasonally flooded forests in the Anavilhanas (Supplementary Figs. 4 and 5) and Tabuleiro do Embaubal (Supplementary Figs. 6, 7, and 8) archipelagos have varied formation ages ranging from hundred to thousand years, but the growth of both archipelagos started during the middle Holocene. These two archipelagos are in the head of lake-like flooded valleys (*rias*) of clearwater and blackwater tributaries of the Solimões-Amazon River. In these settings, flow expansion due to channel widening leads to decreasing water velocity and net sediment deposition. Indeed, the expansion of both archipelagos since 4 ka (Supplementary Figs. 4 and 6) correlates with the high precipitation over the headlands of the Solimões-Amazon River[44,45]. The synchronous response of sediment accumulation in flooded tributary valleys throughout the basin would result from the erosion base level control performed by the backwater effect of the Solimões-Amazon main stem into the tributaries[56,57]. Hence, we propose that sediment accumulation fueling the synchronous expansion of seasonally flooded environments in lowland Amazonia during the Holocene is a combined effect of the sea-level highstand and orbitally-driven strengthening of the SAMS.

In summary, periods of increased precipitation enhance sediment input to river valleys and promote aggradation of floodplains if the erosion base level rises across lowland Amazonia. Sea level would control the erosion base level in the low elevation reach of the Solimões-Amazon River main stem. Sedimentary substrates of floodplains within the Solimões-Amazon River valley near the locality of Tefé (Figs. 1 and 2), situated more than 1800 km upstream the coastline, have elevation below 35 m (high water stage level in Fig. 6b). Thus, the reduced slope of the Solimões-Amazon River in the lowland would allow the long-distance upstream propagation of the backwater effect provoked by long-lasting millennial high precipitation periods occurring during sea-level highstand phases. In this case, sediments transferred from headwater areas during higher precipitation phases accumulate in river valleys and sustain the expansion of seasonally flooded substrates across the lowlands. The condition combining high water discharge in the trunk river and sea-level highstand occurred during the Holocene (Fig. 7b, c), which would explain the synchronous expansion of floodplains and archipelagos in the Amazonian lowlands. On the other hand, higher rainfall during a falling sea level condition would induce channel incision and abandonment of floodplains, which are converted into upland terraces. The strengthening of the SAMS under falling sea level occurred for example from 80 ka to 21 ka[44,58] and could have induced channel incision leading to formation of elevated terraces bounding the Solimões River (22, Fig. 6b, c).

### Response of bird populations to the availability of seasonally flooded habitats

The increase in the availability of substrates that sustain seasonally flooded habitats affects their specialized biota[21]. The riverine landscape scenario reconstructed for lowland Amazonia indicates that the current distribution of seasonally flooded habitats results from significant expansion of floodplains during the Holocene (Figs. 7a and 8a),

suggesting periods of restricted habitat availability and/or reduced connectivity before this recent expansion. Previous comparisons between floodplain and upland forests avifauna have already shown that upland *terra firme* lineages are structured, forming distinct clades in different areas of endemism bounded by large Amazonian rivers, each with its own demographic history, while floodplain lineages are often less structured and thus generally less diverse[59]. Several upland *terra firme* bird populations show a signal of demographic expansion during the Pleistocene, but the trends vary geographically and in timing for the different species[4]. We consistently found a congruent pattern of recent and steep expansion for the species associated with early successional vegetation habitats. Our results clarify the abiotic mechanisms that have probably controlled population connectivity along the Amazonian floodplains through time.

The most recent period of bird populations expansion covers the last 10 ka (Fig. 8b, c), which coincides with the human occupation of Amazonia[60]. Thus, it could be hypothesized that prehistoric human usage of floodplains favored early successional habitat specialists. However, records of prehistoric human occupation across Amazonia are concentrated in upland *terra firme*. Furthermore, human-mediated landscape changes during the early and middle Holocene would have mostly local effects in bird communities, while the regional changes in fluvial sedimentation driven by precipitation and sea level variations could better account for the wide geographical range of the studied bird populations.

Demographic growth observed in birds from floodplain forests during the last 200 ka (Fig. 8c and Supplementary Fig. 9) agrees with the older expansion phases of floodplains in central and western Amazonia[22], which are at least in part coincident with a period of sea-level highstand[42] and strengthening of the SAMS[44]. Enhanced SAMS phases coincident with high sea level occurred for example around 90–150 ka (Fig. 8d, e). However, species associated with the young successional vegetation mostly present on river islands show a distinct demographic pattern. Instead of a sustained population expansion in the last 200 ka, they only show a single recent signal of population expansion (Fig. 8b). This may be explained by the spatially limited habitat available for these specialists in early successional vegetation, as they are restricted to active fluvial channels, mostly islands, resulting in smaller and less genetically diverse populations[21]. Habitat availability for these specialists was favored during the Holocene, when higher sediment accumulation in river valleys sustained more continuous formation of sediment bars. However, early successional vegetation habitats have limited growth space and are ephemeral (decadal-centennial scales) compared to wide and long-lasting (millennial scale) mature seasonally flooded forests, implying a changing environment. Thus, species that occupy wide floodplains with mature forests at large rivers' valleys may have sustained expansion or retraction through time, avoiding local extinctions and maintaining connectivity, because the development of continuous seasonally flooded substrates bounding river channels depends on the sediment accumulation-erosion balance operating at longer time-scales.

The connectivity of seasonally flooded habitats across the Amazon basin depends on the availability of floodplains along the main stem of the Solimões-Amazon River. However, bird community composition differs between eastern and western Amazonia[19], and genomic data suggest recent introgression between closely related populations in central Amazonia[21,61]. Evidence for distinct populations of island birds in the Negro and Solimões Rivers also indicates reduced connectivity across central Amazonia in the past[62]. This evidence, combined with our demographic results, points to a history of changing connectivity of seasonally flooded habitats in central Amazonia, suggesting that the current connectivity may have been recently attained. Space available for the development of floodplains can also increase with valley widening during phases of stable erosion base level, especially in areas such as western lowland Amazonia, where

valleys are flanked by Pleistocene soft sedimentary terrains[22]. This condition would provide larger and more resilient floodplains in western than in central Amazonia, where river valleys are narrower due to the higher resistance of the rocky substrates[63]. Therefore, central Amazonia is susceptible to faster floodplain loss if the Solimões-Amazon River main stem shifts to an erosional regime.

We conclude that the coincidence between the strengthening of the SAMS and sea-level highstand promoted the last expansion of Amazonian flooded environments and their specialized bird populations, which responded differently depending on their habitat association. It is important to note that these last events of bird populations expansion that we characterize here were preceded by a long history of occupation of seasonally flooded habitats by these lineages, with the stem ages of the nine species studied here ranging from about 1 to 6 Ma (Supplementary Data 4). The sister groups of five among the nine studied species occur outside Amazonia, and the sister groups of the remaining four species also occupy flooded environments within Amazonia (Supplementary Data 4). This indicates that specialization to seasonally flooded habitats is older than the populational processes that generated current distributions of birds along the floodplains. Thus, the demographic shift associated to the reconstructed changes in Amazonian fluvial landscape is ongoing. This relationship between habitat association and landscape change may provide crucial information for estimating the response of the seasonally flooded Amazonian ecosystems to future climate change under the current and long-lasting sea-level highstand condition.

## Floodplains extent and atmospheric methane concentration

The expansion or retraction of floodplains of large tropical rivers can also influence the atmospheric concentration of greenhouse gases since flooded environments are major sources of methane ($CH_4$) to the atmosphere. Most of the global $CH_4$ budget variability during the Quaternary has been associated to variations in wetlands extent during glacial (wetlands decrease) and interglacial (wetlands increase) periods[64]. Superimposed on this long-term pattern, changes in the atmospheric $CH_4$ levels follow the precession cycle (Fig. 8f, g), with concentration maxima broadly coinciding with peaks in summer insolation at 30°N[65]. However, ice-core records show a departure from this pattern starting in the middle Holocene, with an atypical rise of around 150 p.p.b.v. in $CH_4$ levels during a period of weak Northern Hemisphere summer insolation (i.e., the late Holocene)[66,67]. There is at present no consensus on whether this rise in $CH_4$ concentration was anthropogenic, mainly linked to rice paddies in eastern Asia (e.g.,[68]), or caused by changes in natural sources (e.g.,[69]).

Our results show that the unexpected global $CH_4$ rise after 5 ka coincides with the expansion of seasonally flooded environments in Amazonia (Fig. 7a, d). This region currently emits roughly $46.2 \pm 10.3$ Tg of $CH_4$ per year, which corresponds to around 8% of global $CH_4$ flux to the atmosphere[24]. Since the vast majority of these emissions are directly linked to flooded areas (e.g., aquatic surfaces, floodplain trees, and macrophytes, 25), the expansion of seasonally flooded habitats proposed in this work may have played an important role on global $CH_4$ budgets. This scenario is consistent with model simulations[69] and ice-core records[70,71] that suggest southern tropical wetlands as the most likely source for the late Holocene $CH_4$ upsurge. However, specific studies are needed to confidently test the connection between the expansion of Amazonian floodplains and global $CH_4$ budgets.

## Limitations and future perspectives

Despite our confidence in the timing of the expansion of seasonally flooded habitats across lowland Amazonia, we cannot exclude the possibility of a slight bias in the geochronological dataset assembled here. This is due to the different preservation potential of sediment bars. Stratigraphic preservation bias is also expressed by the decrease in sedimentation rates with increasing timespan (Supplementary

Fig. 3) that could favor age clusters toward younger deposits due to continuous reworking of sediment bars by dynamically shifting channels. However, the Holocene expansion of floodplains along thousands of kilometers of the Solimões-Amazon River and its tributaries and the age difference between the active floodplains and their bounding elevated terraces (Fig. 6) representing paleo-floodplains[22,63] point out to a synchronous response of the lowland fluvial landscape in the millennial timescale. Notwithstanding these limitations, a similar pattern of late Pleistocene channel incision followed by Holocene expansion of floodplains was also documented in other large South American tropical rivers, such as the Cuiabá megafan draining the Pantanal wetland[72] and the Paraná River[73], where the SAMS also controls the hydroclimate.

Thus, seasonally flooded environments associated to other Neotropical large rivers and, consequently, their specialized biota would also have experienced synchronous responses. Nevertheless, setting up a common timescale for sediment deposition, precipitation changes, and bird population dynamics is still challenging due to different age uncertainties of dating methods applied to these distinct disciplines. Particularly, estimating precise ages of population expansion events is still limited by the uncertainties regarding molecular evolutionary rates and generation times. Our demographic inferences consistently show patterns of population expansion for all taxa and a distinct timing of these events for birds with particular habitat affinities. Upcoming studies that improve our knowledge about genome-wide evolutionary rates will further strengthen the predictive power of phylogenetic and phylogeographic inferences to test geological hypotheses[74,75].

The recurrent expansion or retraction of seasonally flooded environments due to changes in the riverine sediment budget is a potential mechanism promoting biotic diversification and contributing to the high tropical biodiversity. The general character of this mechanism can be tested through studies in other large tropical rivers comparing age-constrained paleogeography of flooded environments to genomic patterns of their specialized biota. Besides climate and sea level controls, we highlight that the sedimentary budget of large tropical rivers has been significantly altered by increasing land use and channel impoundment to build hydropower reservoirs[76,77]. These anthropogenic activities imposed decadal to secular changes in the sedimentary budget comparable to natural millennial-scale changes[76], which can influence the global climate through shifts in natural $CH_4$ emissions. We suggest that synergies between climate change and anthropogenic stressors on the sedimentary budget of large tropical rivers are major threats to the conservation of the seasonally flooded environments and their unique specialized biota.

## Methods
### Geochronology
Sediment samples for OSL and radiocarbon dating were collected during field surveys performed between 2011 and 2015. Samples were retrieved during the low water season (September–November) in outcrops along channel margins, pits (0.5–1 m depth) or through coring (up to ~10 m depth) using a manual auger adapted to collect samples using opaque plastic liners. Samples were collected in terraces along channel margins or islands covered by *várzea* or *igapó* forests. The studied deposits were described using a sedimentary facies analysis approach. They consisted of a lower sand layer covered by upper muddy layers, sometimes with thin intercalations of sandy and muddy layers, representing a sedimentary succession of channel sandy bar stabilization. Sediment samples were retrieved at the top of the lower sandy layer to determine the initial time of bar stabilization in specific sites or along vertical profiles to constrain sediment aggradation's chronology. Samples for OSL dating were collected using opaque plastic or aluminum tubes to avoid light exposure. Radiocarbon dating through Accelerator Mass Spectrometry was applied in wood and charcoal fragments occasionally found within sediment layers. The OSL dating procedures were carried out in the Luminescence and Gamma Spectrometry Laboratory of the Institute of Geosciences of the University of São Paulo. OSL dating was applied in multigrain aliquots of quartz (sand fraction) or polymineral aliquots (silt fraction). The preparation of quartz or polymineral concentrates for the determination of equivalent radiation doses included wet sieving to isolate the <63, 63–125, 125–180, or 180–250 μm grain size fractions (Supplementary Data 1), depending on the sample texture which ranged from silt to sand. Fine silt (4–11 μm) was used for OSL dating of muddy samples without significant amounts of sand. The fraction finer than 63 μm was used to separate the 4–11 μm through settling in distilled water and centrifugation. Treatments with $H_2O_2$ and HCl 10% were applied to eliminate organic matter and carbonates, respectively. Quartz sand grains were separated from heavy minerals and feldspar grains through heavy liquid solutions (lithium metatungstate) at densities of 2.75 and 2.62 g/cm³, respectively. The heavy liquid separation was not performed for samples dated using the fine silt fraction (4–11 μm). Quartz concentrates in the sand fraction (63–125, 125–180, or 180–250 μm) were submitted to HF 38% treatment for 40 min to etch the outer rind of grains damaged by alpha particles followed by another HCl treatment to eliminate eventually precipitated compounds and wet sieving to keep grains in the target grain size interval. Luminescence measurements for equivalent dose determination were carried out in two Risø TL/OSL DA-20 readers (Risø National Laboratory) equipped with built-in beta radiation sources ($Sr^{90}/Y^{90}$), with dose rates ranging from 0.075 to 0.112 Gy/s, blue and infrared LEDs for stimulation and Hoya U-340 filters for light detection in the ultraviolet band. For pure quartz aliquots (sand fraction), equivalent doses were estimated using the single-aliquot regenerative (SAR) dose protocol[78]. Some muddy samples (fine silt fraction) from riverbanks along the Solimões River did not allow the extraction of pure aliquots of quartz. Thus, luminescence dating of these samples was performed using polymineral aliquots (quartz and feldspar). Then, equivalent doses were estimated using the post-infrared blue stimulated luminescence signal[79]. This approach aimed to measure the OSL signal from quartz after depletion of the feldspar signal. Ages obtained through polymineral aliquots were cross-checked through comparison with ages of pure quartz aliquots from sandy layers occurring in the same profile. The protocols used for equivalent dose estimation are summarized in Supplementary Material (Supplementary Table 1). The suitability of the mentioned SAR protocols in estimated laboratory known doses was appraised through dose recovery tests considering the natural dose range of the studied sediment samples. Calculated-to-given dose ratios were within 0.9–1.1. Only aliquots with recycling ratio within the 0.90–1.10 interval and recuperation less than 5% were used for equivalent dose calculation through the Central Age Model[80]. Dose rates (Supplementary Data 1) were calculated through radionuclides ($U^{238}$, $Th^{232}$, and $K^{40}$) concentrations determined using gamma-ray spectrometry with a high purity germanium detector, with an energy resolution of 2.1 keV and relative efficiency of 55%, encased in an ultralow background shield (Canberra Instruments). Radionuclides concentrations were converted to dose rates using conversion factors[81]. The contribution of cosmic radiation to dose rate was calculated through sample depth, elevation, latitude, and longitude[82]. Besides the sediment burial ages obtained in this study, we compiled OSL and radiocarbon ages from the literature. Thus, our geochronological dataset to constrain sediment deposition in the floodplains and archipelagos of the Amazonian rivers include 93 OSL ages and 23 radiocarbon ages obtained in this study, and 53 OSL and radiocarbon ages from published works. Supplementary Data 1 summarizes the OSL and radiocarbon age data from seasonally flooded substrates obtained in this study and compiled from the literature. OSL and radiocarbon ages from higher elevation terrains bounding seasonally flooded areas were also compiled from the literature (Supplementary Data 2).

## Bird genomic data sampling and historical demographic analyses

For each of the nine studied species, samples were selected aiming to cover most of their known geographic distribution, spanning seasonally flooded habitats throughout Amazonia (sampling details in Supplementary Data 4 and maps in Supplementary Fig. 2). DNA was extracted using QIAGEN DNeasy Blood and Tissue kit and DNA extracts were sent to Rapid Genomics for sequence capture using a probe set targeting 2321 UCEs[59]. Sequences from each species were processed independently from raw reads to final UCE alignments through the Phyluce pipeline[83]. First, using the *illumiprocessor* function, raw reads were cleaned from adapter contamination. Second, a single sample with high amounts of reads was assembled into contigs using SPAdes[84]. Contigs were matched to the UCE probes sequences with *phyluce_assembly_match_contigs_to_probes*, and recovered contigs were used as a reference to all other samples. Afterward, each sample's reads were mapped against reference alignments and phased using *phyluce_snp_bwa_multiple_align* and *phyluce_snp_phase_uces*, respectively[85]. Finally, all samples' recovered sequences were aligned into separate UCEs sequences files with MAFFT[86] using *phyluce_align_seqcap_align* and trimmed to improve alignment quality using *phyluce_align_get_gblocks_trimmed_alignments_from_untrimmed*. Samples with low coverage (usually less than 1500 UCEs) were discarded and we selected only UCE alignments including all samples with *phyluce_align_get_only_loci_with_min_taxa* to be used in subsequent analyses.

Besides the UCE datasets, two widely used mitochondrial loci (ND2 and cytb) were recovered to add information from a distinct genomic source and provide additional information for time calibration of demographic reconstructions. The mitochondrial genome is usually recovered as a byproduct of UCE sequencing[87]. For each species, cytb and ND2 reference sequences were downloaded from GenBank, and samples' reads were mapped to references and aligned in Geneious 7.1[88]. In the absence of cytb or ND2 sequences available at GenBank for a species, its closest taxon was used as a reference.

Since population structure and isolation by distance can mislead demographic analyses, three different methods were used to assure that demographic analyses were performed on the best population arrangement. To look for areas of higher or lower gene flow than expected under simple isolation by distance, we modeled the EEMS (Supplementary Fig. 12) of all studied species[89]. For each species, Euclidean genetic distances between samples were estimated from the SNPs data with ADEGENET'S dist.genpop function (Supplementary Fig. 13[90]). The described distribution of a species was used as the distribution polygon, which was further edited in Qgis when necessary to include samples outside the polygon. Depending on the size and complexity of the species' distribution polygon, from 500 to 700 demes were selected. We present a single run of $1 \times 10^7$ generations for each species, excluding the first $5 \times 10^6$ as burn-in. The convergence of the runs was visually assessed in the posterior probability trace plots. Additional tests for some of the studied species with longer chains and different numbers of demes were also run to confirm the consistency of the results.

We used two different classification approaches to estimate the best population arrangement using sNMF and k-means, implemented through LEA and ADEGENET R packages, respectively[90–92]. For both methods, we tested the likelihood of *K* values from 1 to 10, representing the number of populations. In sNMF, different alpha values were compared (1, 10, 100, and 1000), and the best result was chosen by the smallest value of cross-entropy. The k-means classification in ADEGENET is applied with find.clusters() function and infer the number of genetic clusters in our samples comparing the Bayesian Information Criterion of each classification. Both sNMF and k-means yielded similar results, suggesting *K* value of 1 population for all species except for *K. orenocensis*, for which both analyses indicate the existence of 2 distinct genetic groups (Supplementary Fig. 13),

corresponding to the two described subspecies, and only one of them (*K. o. sclateri*) was used in demographic analyses.

Demographic analyses were performed for each taxon independently with EBSP in BEAST 2.5[93]. We followed the approach suggested by Trucchi et al.[94], where they show highly consistent results comparing datasets of 50 nuclear loci with 2, 3, and 4 to 6 SNPs per loci, but used loci with 4 to 6 SNPs for being more informative, and suggested a calibration strategy combining information from nuclear and mtDNA markers. We thus randomly sampled 50 UCE loci with at least four informative sites, plus the two mitochondrial loci for all individuals of each species[94]. Site models were inferred independently for each locus in jModeltest[95]. Site models and clocks were unlinked for all loci and the tree was linked only for the two mitochondrial loci. To avoid over-parameterization, strict clocks were used for estimating the rates. Tree prior for each locus was set as "Coalescent Bayesian Skyline" and ploidy was set as 0.5, 1.5, and 2 for mitochondrial, Z-linked, and auto-somal loci respectively. To identify Z-linked UCEs, they were mapped to a complete Z chromosome of Zebra-finch downloaded from GenBank (Accession: CM020861) using Geneious. A normal distribution clock prior was used for mitochondrial markers to calibrate the demographic reconstructions, with initial and mean values set as 0.0105 and sigma parameter as 0.0034, encompassing the widely used rate of 2% substitution/site/million years for birds' mtDNA[96]. For each UCE, a gamma clock prior was used with alpha set as 0.1256, beta as 1.0, offset as $1 \times 10^{-5}$, and initial value as 0.0025. This setting centers the median, and therefore most of the estimates, around 0.0025 matching the proposed mean substitution rate for the genome of birds of $2.5 \times 10^{-9}$ substitutions per site, per generation, and one year generation time for birds specialized in seasonally flooded Amazonian environments[21,97]. The populationMean.alltrees prior was set as normal with 1.0 as mean and 0.1 as Sigma, and "alltrees" operators were multiplied by the number of loci[98]. Finally, chain length was set as $5 \times 10^8$, storing every $5 \times 10^4$ for tracelog and $1 \times 10^4$ for EBSPlogger. For each demographic reconstruction, ESS values of parameters were checked on Tracer[99]. Following the tracer log, 3 to 5 independent runs were performed for each species and combined with logcombiner removing the first 10% as burnin. For all species, after combining the independent EBSP runs, most parameters reached very high ESS values (>1000). Although some analyses had low ESS values for individual parameters (<100), the observed consistency across different runs and acceptable values for the most important parameters support the convergence of the analyses.

Two additional analyses were performed to further explore the demographic history of the studied species using a larger portion of the UCE dataset. Firstly, demographic models representing population expansion, stability or retraction were compared for each species using an Approximate Bayesian Computation Framework in DILS[35]. To keep most loci and considering the data were already filtered in Phyluce, in data filtering max_N_tolerated was set to 0.3, Lmin to 100 and nMin to the number of sequences per species. The same previously used mutation rate of $2.5 \times 10^{-9}$ substitutions/site/generation with a one year generation time was used to calibrate the analyses. Minimum and maximum time of demographic change were set to 100 and $5 \times 10^5$ generations, respectively, and limits of population sizes were left as the default values of 100 for the minimum and $10^6$ for the maximum. Time of demographic changes was assessed based both on the posterior and optimized posterior probabilities from the Neural Network algorithm.

The second analysis aimed to test for shared responses to environmental changes, which would result in temporally congruent demographic changes. In this analysis, we used the full Bayesian model choice framework applied in *Ecoevolity*[36]. For that, we used the complete phased UCE dataset, concatenated in a single nexus file per species, as recommended[36]. In event_model_prior, a gamma_distribution with shape = 0.7 and scale = 8 was set for the concentration parameter of the Dirichlet process. For our data, this setting results in

similar prior probabilities for most estimates of demographic changing events, reducing prior bias in model choice (Supplementary Fig. 10). For the event_time_prior, a uniform distribution was set with minimum of 1000 and maximum of $5 \times 10^5$ years. For the Markov Chain Monte Carlos, a chain length of $15 \times 10^5$ steps sampling every 1000 samples was defined. Operator settings were left as default. An uninformative uniform prior was set for the population size, with mean, minimum and maximum values of $5 \times 10^5$, 1000 and $10^6$, respectively. The same mutation rate of $2.5 \times 10^{-9}$ substitutions/site/generation was used, with an uniform distribution and minimum and maximum values of $1.5 \times 10^{-9}$ and $3.5 \times 10^{-9}$. Three independent runs were performed and checked for convergence using Tracer.

### Reporting summary

Further information on research design is available in the Nature Research Reporting Summary linked to this article.

## Data availability

All data needed to evaluate the conclusions in the paper are present in the paper and/or in the Supplementary Material. OSL and radiocarbon ages data are available Supplementary Data 1 and Supplementary Data 2. Data used to calculate sedimentation rates are available in Supplementary Data 3. Genomic sampling information is presented in Supplementary Data 4. Genomic data are deposited as raw reads in the National Center for Biotechnology Information Sequence Read Archive (Bioproject PRJNA819080). All input and output files from demographic analyses are deposited at https://www.github.com/eduardoschultz/floodplains_demographies. Maps and graphic representations of genomic and geochronological data, measurement protocol used for OSL dating and additional information on bioinformatics procedures are presented in the Supplementary Information file.

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

## Acknowledgements

Dr. Thays Desirée Mineli and Luciana Nogueira are thanked for the support with laboratory and analytical procedures for luminescence dating. This research was funded by Fundação de Amparo à Pesquisa do Estado de São Paulo (A.O.S received FAPESP grants #2011/06609-1, #2016/02656-9 and #2018/23899-2). Genomic analyses were funded by the PEER/USAID grant to C.C.R. (AID-OAA-A-11-00012). A.O.S., F.N.P., C.H.G., C.M.C., and C.C.R. are supported by CNPq (grants #304727/ 2017-2, #302411/2018-6, #304413/2018-6, 312458/2020-7 and # 311732/ 2020-8). F.N.P. was supported by a postdoctoral fellowship from FAPESP (grant#2014/23334-4) and British Society for Geomorphology - Early Career Researcher Grant. C.M.C. acknowledges the financial support from FAPESP (grants #2018/15123-4 and #2019/24349-9), CAPES (grants #564/2015 and #88881.313535/2019-01), and the Alexander von Humboldt-Stiftung.

## Author contributions

A.O.S., F.N.P. and C.C.R. designed the study. A.O.S., E.D.S., C.C.R. and D.J.B. wrote the manuscript with assistance of F.N.P., C.H.G., I.D.W., C.M.C., F.W.C., and R.P.A. E.D.S., and C.C.R. performed genomic data acquisition and analysis. F.N.P. compiled OSL and radiocarbon age data and participated on field surveys. D.J.B. compiled paleoclimate and sea level dataset and participated on field surveys. D.F.S. performed sampling and OSL dating of sediments from the Tabuleiro do Embaubal in the Xingu River. D.F.C. carried out OSL dating of samples from Anavilhanas in the Negro River. C.E.M. performed OSL dating of samples from the middle Solimões River. M.P.F. was responsible for sampling and OSL dating of samples from the lower Solimões and upper Amazon Rivers. G.H.G. and I.D.W. assisted with planning and execution of field surveys. C.M.C. and F.W.C. contributed with interpretation of paleoclimate records. R.P.A. assisted with interpretations on fluvial sedimentary dynamics. All authors reviewed and approved the final version of the manuscript before submission.

## Competing interests

The authors declare no competing interests.
