## [Peer Review File · Nature Communications]

Rainfall and sea level drove the expansion of seasonally flooded habitats and associated bird populations across AmazoniaREVIEWER COMMENTS

Reviewer #1 (Remarks to the Author):

Sawakuchi et al. compiled new and published data to reconstruct the sedimentary history of floodplains in Amazonia, and summarize the chronology of the formation of seasonally flooded habitats. They further integrated genomic data from nine bird species to show that population expansion occurred in tandem with the formation of river island and floodplain forests. I do not have an expertise in geology, so I will focus my review on the biological components of the study. The study is quite unique and powerful in that it combines sedimentary and genomic data to produce a holistic view on the evolution of the Amazonia landscape and biota. This approach is noteworthy and will be of broad interest. My main comments focus on the details and power of the genetic analyses, and the interpretation of the evolution of the focal lineages over deeper evolutionary time.

First, it would be helpful to include more details about the genetic data and a more definitive test about patterns of population expansion. I would view the skyline analysis as a first pass and one of several approaches to infer demographic history. I would recommend explicitly testing different demographic models for the river island and varzea birds. For example, fastsimcoal2, MOMI2, and PipeMaster would allow the use of all loci and more rigorously test alternative demographic models. The program multi-DICE allows for explicit testing of simultaneous expansion, which is relevant for the narrative of the study. My concern with just using a skyline analysis, which is really informative, is that the interpretation is often descriptive (i.e., the line goes, up, down, or stays flat). In contrast, estimating the probabilities of alternative demographic models would clarify how much support there is for particular scenarios. Two recent demographic studies on Amazonian lizards (Prates et al. 2016 PNAS) and birds (Thom et al. 2020 Sci Adv) have used these types of approaches.

Second, my other concern is that it is unclear what is driving the different demographic signatures between the two groups. It would be helpful to report summary stats of the data including sample sizes, variation, missing SNPs, Tajima's D, etc. to show that the patterns are robust to artifacts in the data.

Prates, et al. (2016). Inferring responses to climate dynamics from historical demography in neotropical forest lizards. *Proceedings of the National Academy of Sciences*, 113(29), 7978-7985.

Thom, G., et al. (2020). Quaternary climate changes as speciation drivers in the Amazon floodplains. *Science advances*, 6(11), eaax4718.

An important part of the narrative that is not addressed is the deeper evolutionary history of birds that specialize in seasonally flooded forests. In order to calibrate their population history to sediment changes then you would also want to know when these lineages colonized these habitats. I suspect these taxa are on large branches, so what can be said about their demographic histories prior to 11,000 years ago? The long flat lines in the skyline plots likely reflect a lack of information in the genetic markers and not long periods of evolutionary stability.

Parallel expansion in river island birds is clearer to understand than in the floodplain forest birds. The floodplain forest birds have what appears to be low/gradual expansion. I agree the two groups of birds appear to show different patterns but the coupling of floodplain expansion and bird expansion is less apparent. For example, if floodplain birds showed a distinct pattern from terra firme birds then there would be more support for the narrative. The lack of resolution in the genetic data and skyline analysis, and not providing context of the deeper evolutionary history of the lineages limits what can actually be said.

Additional comments

Line 58. "Other authors interpret that landscape change had a minor role in Amazonian diversification and propose that most diversification was driven by long-distance dispersal or intrinsic biotic processes over a relatively stable landscape"

This sentence is a mischaracterization of what is said in reference 9. The authors never said the landscape was stable. I don't know of anyone that has ever said the Neotropical landscape was stable. There is clear evidence the landscape is not even stable on a temporary time scale. I also wouldn't use the words most diversification because the authors only examined a small proportion of taxa. They tested two models and there was no support for the genetic patterns predicted from congruent evolution with earth history. From this text they proposed a model for diversification.

The sentences from 59-63 need to be adjusted accordingly.

There also seems to be confusion, not particular to this paper, by what is meant by tropical stability. It is a statement relative to temperate regions, which were covered by mile thick ice sheets. So the tropics were more stable than the temperate region. This does not mean that the tropical landscape and climate did not change dramatically. I am not sure if this is part of the basis for using the word stable in this paper.

Line 59. "However, both views are limited by the lack of chronological information (i) to constrain the presumed environmental changes that split populations or (ii) to support landscape stability during the diversification of specific taxa."

I agree chronological information is useful, but it could also be really misleading if the population splits do not have anything to do with particular environmental changes.

And same as before, ref 9 does not say the landscape was stable. The word stable is not used in that manuscript (I just checked!). The study found that diversification was random with respect to landscape changes or occurred after the formation of the rivers and Andean uplift. The distinction is important.

Line 146. It says UCE loci were used but in the Methods it says UCEs and two mtDNA were used. Were different sets of analyses done?

Discussion. Without tabs to start new paragraphs, it is harder to follow the start of new paragraphs.

Line 177. What is meant by an opposite population dynamic is expected for seasonally flooded and upland forest species? Their population would have contracted? The demographic analyses I have seen from Lais Coelho's published dissertation shows concordant demographic expansion in upland forest birds.

Line 261. "suggest recent contact of closely related populations in central Amazonia"

Do you mean secondary contact and introgression? The sentence is unclear.

Line 263. "This evidence, combined with our demographic results, points to a dynamic history of changing connectivity of seasonally flooded environments in central Amazonia, suggesting that the current connectivity may have been recently attained."

Isn't the expectation that river island habitats are ephemeral and that the rivers shift their courses. This is observable on Google Earth.

Line 388. "Afterward, each sample's reads were mapped against reference alignments and phased (67)."

Include more details. This explanation of how the data was phased is not repeatable. The same could be said for the entire bioinformatic sections. It is too brief to be repeatable. Is there a more detailed explanation in supplementary material?

Line 394. If you have the entire mt genomes why use just ND2 and cyt b? There are decent references available.

Line 399. In addition to checking for population structure, a test for isolation-by-distance is also warranted because IBD could also bias the demographic modeling. For example, species with IBD may show false positives of population expansion.

Line 412. What is the justification of requiring at least 4 informative sites? In order to avoid ascertainment bias, loci should be randomly selected. Very different demographic signatures can be inferred depending how loci subsampled. For example, preferentially selecting more variable loci is likely to exclude loci that do not show an expansion.

The other issue with subsampling, which I assume was done because BEAST could not handle the size of your data, is that if it is only done once, it is unclear how much variance there would be across different subsamples. It is important to show a particular pattern is robust to a sampling procedure.

The images and figures are fantastic. It might help to include a silhouette of one of the focal species on the skyline plots, so the genetic analysis stands out.

Figures 4 and 5. The long tails of the right side of the skyline plot are not informative of the demographic history for the deeper (or lack thereof) coalescent events. The data has its limits. The legends do not explain to the reader how to interpret the skylines plots.

**Brian T. Smith
American Museum of Natural History**

Reviewer #2 (Remarks to the Author):

I am an Ornithologist, and thus not qualified to comment on the geochemical analyses in this paper.

That being said, I found the overarching story of the manuscript compelling, novel, and as a much-needed multidisciplinary perspective into biogeography; and an important contribution to our understanding of how populations interact with earth's history.

The detailed analysis of the geochemical history of the Amazon basin provides a novel context for understanding demographic history, and thus biography of amazonian birds. Biogeography has a long tradition of heuristic methods and inference, and so detailed and objective comparisons like this are consequential.

I believe this work does support the conclusions, although I have some comments about how the conclusions could have been more compelling (see below). The methods appear to be sound, from the biological perspective. I cannot comment on the geochemical analysis.

I only have a few general comments.

The first comment is that anthropogenic effects were not discussed. There is recent evidence that people may have drastically affected the Amazon up to ~10,000 years ago,

which is within the timeline for expansion of some of these species, and human-mediated habitat change often results in edge and early-succession habitats that are favored by these species. I think this manuscript would be stronger if the authors consider the possibility of the role of anthropogenic effects, even if subjectively, or acknowledged this as a caveat.

My second comment refers to biological sampling. I think that this study could have been strengthened by including a "control" group of upland avifauna. The paper mentions an explicit hypothesis that upland avifauna should show opposite patterns to island and floodplain avifauna, and the inclusion of these species would have the potential to provide a much more compelling story. As it is, many of the species show fairly similar demographic histories, and so the correlations may be conditionally independent on some other factor; for example a history of reticulate diversification and extirpation, independent of habitat and ecology. Separate species with similar distributions but that do not rely on these floodplain and island habitat showing drastically different demographic histories would greatly strengthen the inference in this analysis.

I also think a population phylogenetic approach could have helped this analysis, especially for island taxa. This could have provided diversification and demographic information within a larger context. For example, all island species show recent population expansions, and the authors mention this could have been due to increased connectivity of island areas. If these species were restricted to few and distant regions of islands, we would expect divergence times to coincide with this period, especially among species that show divergence within island taxa. Or, divergence from sister species not occupying islands may have occurred during previous periods of extensive river island habitat. Including within-population diversification as well as timing of diversification from non-island sister species could provide much insight into the demographic history of these species.

Reviewer #3 (Remarks to the Author):

This is a very interesting study that reports a series of dates from Amazonia in an attempt to assess the development of seasonally flooded habitats and the connection between flooded habitats and the evolution of endemic bird species. The study also assess the connection between regional climate, global sea level, the global carbon cycle and sedimentation in Amazonia.

The results of this work are interesting and provide new and important insights into the connection between the sedimentary budgets of Amazonian lowland floodplains and the diversification of Amazonian bird species. The work also draws potential connections between the significant expansion of floodplain environments and a global increase in methane beginning at ~ 5 ka. Finally, the work connects sedimentary budget in Amazonia to both sea level and regional climate in a compelling way. The work has significant implications for understanding the development of the Amazonian basin, the diversification of its endemic taxa, and the connection between Amazonian sedimentary environments and the global carbon cycle, which in turns has potential implications for current and future climate change.

The work does a reasonably good job supporting its conclusions; however, the results could be laid out in a more compelling and convincing way with some revisions. In particular, the geochronology section would be improved if the following issues were addressed in revision:

-Introduction of the terrace environments and their dates earlier in the geochronology results section, rather than in the discussion of bird demographics. Although these dates are compiled from the literature, the sort of come out of nowhere when first introduced.

- The explanation of rates of deposition, and in particular about "faster island formation" is not shown in a compelling way, partially due to the way the figures are illustrated and partially because not all of the information is really presented. I suggest that you could present this in a compelling way by showing some sediment thickness vs. dates in a figure and/or in a table showing sediment accumulation rates (and uncertainties).

- The stratigraphic figures are important, however, in their current form, Figures S5 and S6 are not very useful because there is no way to assess elevation/topographic/stratigraphic similarities or differences between the sections. The sections need to be shown relative to a datum to compare the sections and the dates.

- It would also be useful to see some photos of what the deposits and the facies look like in situ.

-Figure 4, which is where the geochronology data is shown could be improved for clarity and to better emphasize the results of the work in several ways (1) panel 4B seems important, but I'm uncertain what its showing. Is this density of floodplain islands? If so, how was this determined? (2) In panel 4A, it is difficult to resolve different symbols and colors. Why two colors for each? (3) in panel 4A, what is the spread of dates on the y-axis indicating? (4) in panel 4F, add the location of each cave (i.e., eastern, northern, etc) to help clarify part of the discussion section.

-Figure 5 would benefit from the addition of a sea level curve if possible.

Most of the discussion section is well reasoned, but I was wondering about the comment about episodic sedimentation increasing at 4 ka on line 209, because in the Tabuliero do Embaubal section it looks in some sections at 4 ka and then some between 2 and 3 ka?

Finally, in the discussion section, the potential connection between methane concentrations and flood plains is a really interesting idea. I encourage the authors the develop this idea a bit more because it would definitely increase the significance of the work.

The methodology is sound and reasonable presented. The methods section would benefit a bit from some discussion about how the vertical profiles were collected (i.e., cores, sediment pits, something else?).

-Figure S3

Reviewer #4 (Remarks to the Author):

As a geologist that works on rivers and their evolution, I will confine my comments to the geological part of the paper, rather than comment on the more biological aspects. The latter are very interesting, but I know nothing about them.

With that said, as a geologist that works on rivers, I have a hard time assessing what the authors have done. Granted, this is a tough environment to work in, but I think a lot of the fluvial part of their story is overinterpreted relative to documentation of their story. I have no issue with the coupling of, or separation of, the effects of sea-level change and climate on river behavior, both types of drivers are inherently related over the time scales of their study. But I have a difficult time evaluating the stratigraphic context and significance of their geochron data, and the reasonableness of their interpretations about rates and scales of fluvial processes over time. I therefore cannot fully evaluate their story, and therefore cannot recommend the paper for publication without major revisions.

Here are a few specific comments:

121-135 – I am uncomfortable with inferences about rates from a data set like this. This paragraph infers things like “highly intensified”, a “retreat phase”, “faster island formation”, and “.....show depositional and colonization rates varying.....”. I am not really clear how they arrive at these rate-based interpretations, other than the sheer numbers of ages for specific time periods. The enormous and impressive number of OSL and 14C ages is, in itself, an important dataset, but it’s hard to see the context within which they can be interpreted in terms of rate-like statements.

179-194 - It is almost a certainty that the entire valley and the channel were at a significantly lower elevation for much of the glacial-period sea-level lowstand, even this far upstream, and especially during the LGM. An interesting paper by Mertes and Dunne (I think), speculates on the lowstand long profile of the Amazon, which is consistent with the well-defined lowstand long profile of the Mississippi. The incision tapers upstream, of course (as the authors recognize), but it can extend a long ways, and I suspect, as the authors discuss, this area is still in the part of the system that aggraded a lot during the latest Pleistocene and early Holocene period of rapid sea-level rise, such that any records of flood plains from that time would be buried, or at the very least, the record of those times is biased against preservation and discovery using the methods they use. Moreover, most rivers have several terraces that represent that period of incision, such that the incised valley topography has lots of variability, the glacial period flood plain would have been inset into and confined by those terraces. The broad flood plain of the late Holocene developed over the top of, and therefore obscures, that variability because it is now only confined by higher valley walls.

206-210 – I do not see how such interpretations come from this type of dataset. What if they were able to date every layer, would things still be interpreted as episodic? Over what time scale does something become episodic - decades, centuries, millennia? And when one bar was not active, did another bar somewhere else become more active?

295-300 – Absolutely this dataset is biased by differential preservation, and the ensemble of ages they cite do reflect a bias towards younger ages - that’s the normal case. One thing that would be useful in this respect would be to actually map the terraces and bars that were active at different times, and therefore have weaker or stronger development of forest – maybe by millennia or something, or even plot samples locations by age – different colors for each millennium or something. One can then evaluate the spatial scales of preservation. But right now, I see no way to evaluate changes through time because the geochron is not really tied to an independently developed stratigraphic model. Instead, they have a large number of ages, but it’s difficult to see what they actually mean. In any migrating river, preservation of a bar and overbank fines is tough due to constant migration and reworking. The surprise would be if something else would have emerged, not that this common bias towards preservation of younger strata is there.

Figures 1-3 are low res DEMs and satellite images that are not really that informative but take a lot of space that might otherwise be used for data or analyses. Moreover, generally, the figure captions need more detail.

Figure 4 – 4A is not an effective way to plot these data. I am not sure what 4B is, from the caption I was expecting radiocarbon ages. But they are included in 4A. And the Y-axis needs to have units, I have no idea with regards to B what density refers to?

Figure 5 – I am not sure what to make of this figure, they are arguing for large-scale correlation and causality, but it’s hard to understand this from the text, and the data, and it therefore seems like sort of an eyeball or qualitative trend matching test, not anything rigorous that shows cause and effect. This was the norm a while ago, but not so much now.

SPECIFIC RESPONSES TO REVIEWER COMMENTS

***Reviewer comments are in italics**

Reviewer #1 (Remarks to the Author):

R#1-BSmith: Sawakuchi et al. compiled new and published data to reconstruct the sedimentary history of floodplains in Amazonia, and summarize the chronology of the

formation of seasonally flooded habitats. They further integrated genomic data from nine bird species to show that population expansion occurred in tandem with the formation of river island and floodplain forests. I do not have an expertise in geology, so I will focus my review on the biological components of the study. The study is quite unique and powerful in that it combines sedimentary and genomic data to produce a holistic view on the evolution of the Amazonia landscape and biota. This approach is noteworthy and will be of broad interest. My main comments focus on the details and power of the genetic analyses, and the interpretation of the evolution of the focal lineages over deeper evolutionary time.

First, it would be helpful to include more details about the genetic data and a more definitive test about patterns of population expansion. I would view the skyline analysis as a first pass and one of several approaches to infer demographic history. I would recommend explicitly testing different demographic models for the river island and varzea birds. For example, *fastsimcoal2*, *MOMI2*, and *PipeMaster* would allow the use of all loci and more rigorously test alternative demographic models. The program *multi-DICE* allows for explicit testing of simultaneous expansion, which is relevant for the narrative of the study. My concern with just using a skyline analysis, which is really informative, is that the interpretation is often descriptive (i.e., the line goes, up, down, or stays flat). In contrast, estimating the probabilities of alternative demographic models would clarify how much support there is for particular scenarios. Two recent demographic studies on Amazonian lizards (Prates et al. 2016 PNAS) and birds (Thom et al. 2020 Sci Adv) have used these types of approaches.

Response: Following the reviewer suggestion, two additional demographic analyses were included to complement EBSP results. In DILS (Fraïsse et al., 2021), the comparison between demographic models using hABC also recovers demographic expansion for all species with high posterior probabilities (>0.98) and distinct dates of expansion for species with distinct habitat preferences. We also tested for simultaneous expansions with *Ecoevolity* (Oaks, 2019), which suggests a single event of recent expansion for the island specialists (*C. vulpecula*, slightly more recent), while floodplain forest taxa expanded independently and at older time frames. Methods were updated (lines 457-551). The results of these new analyses were included in the text (lines 177-199) and in the supplementary material (figs. S10-14, table S4 and "SuppMaterial_Bioinformatics). We also note that EBSP analyses, although having a descriptive interpretation, had excluded the possibility of no demographic change.

R#1-BSmith: Second, my other concern is that it is unclear what is driving the different demographic signatures between the two groups. It would be helpful to report summary stats of the data including sample sizes, variation, missing SNPs, Tajima's *D*, etc. to show that the patterns are robust to artifacts in the data.

Response: The information was included in table S4. Missing data was not allowed in SNPs dataset.

R#1-BSmith: Prates et al. (2016). *Inferring responses to climate dynamics from historical demography in neotropical forest lizards. Proceedings of the National Academy of Sciences*, 113(29), 7978-7985.

Thom et al. (2020). *Quaternary climate changes as speciation drivers in the Amazon floodplains. Science advances*, 6(11), eaax4718.

An important part of the narrative that is not addressed is the deeper evolutionary history of birds that specialize in seasonally flooded forests. In order to calibrate their population history to sediment changes then you would also want to know when these lineages colonized these habitats. I suspect these taxa are on large branches, so what can be said about their demographic histories prior to 11,000 years ago? The long flat lines in the skyline plots likely reflect a lack of information in the genetic markers and not long periods of evolutionary stability.

Response: Yes, we agree with the interpretation. Information on sister species/sister clade was included in table S4 for all species studied here, including their preferred habitat, their distribution, and the age of divergence in relation to the studied species. We also agree that the population expansions described here correspond to the most recent population change that generated current distributions and genomic diversity. It is possible that past changes in availability of floodplains habitats caused earlier cycles of contraction and expansion of these populations that we are not able to reconstruct. Nevertheless, the fact that the last expansion differs consistently between taxa that use the distinct habitats indicates that populations are tracking their preferred habitats, which is the main information that genomic analyses are contributing to understand how the landscape history is affecting birds populations. We have added this to the discussion section (lines 270-282 and 325-336).

R#1-BSmith: *Parallel expansion in river island birds is clearer to understand than in the floodplain forest birds. The floodplain forest birds have what appears to be low/gradual expansion. I agree the two groups of birds appear to show different patterns but the coupling of floodplain expansion and bird expansion is less apparent. For example, if floodplain birds showed a distinct pattern from terra firme birds then there would be more support for the narrative.*

Response: This is a very good point. Many terra firme birds also show signal of population expansion, but not as recent as the signal found here for floodplain specialist birds. We have included discussion about this issue in the text (lines 274-282).

R#1-BSmith: *The lack of resolution in the genetic data and skyline analysis, and not providing context of the deeper evolutionary history of the lineages limits what can actually be said.*

Response: We have now added the information on deeper evolutionary history and performed new analyses that improve the resolution of comparisons of genomic data (see new results section "Demography of bird populations" - lines 177-199 - and cited supplementary materials). The new analyses confirm distinct demographic histories for the two groups of species, with more recent demographic expansion for the species associated to island and open environments, even though all studied lineages originated more than 1 Ma ago and thus went through the landscape changes during a longer time span (Middle-Late Pleistocene).

R#1-BSmith: *Additional comments*

Line 58. "Other authors interpret that landscape change had a minor role in Amazonian diversification and propose that most diversification was driven by long-distance dispersal or intrinsic biotic processes over a relatively stable landscape"

This sentence is a mischaracterization of what is said in reference 9. The authors never said the landscape was stable. I don't know of anyone that has ever said the Neotropical landscape was stable. There is clear evidence the landscape is not even stable on a temporary time scale. I also wouldn't use the words most diversification because the authors only examined a small proportion of taxa. They tested two models and there was no support for the genetic patterns predicted from congruent evolution with earth history. From this text they proposed a model for diversification.

The sentences from 59-63 need to be adjusted accordingly.

Response: We have modified the text (lines 57-60) to correct any misinterpretations related to Smith et al. (2014). Our idea here was to emphasize that the riverine landscape (floodplains and their conversion into uplands) has changed a lot and in diverse ways during the Quaternary, with distinct habitat types experiencing distinct histories. This suggests that incongruence is actually expected, depending on habitat use and ecological characteristics of each species. The idea was to broaden the scope of Amazonian biogeography to avoid only looking for congruence and include more nuances in the investigation about the relationship between landscape history and biotic diversification (see revised discussion section on lines 270-336).

R#1-BSmith: *There also seems to be confusion, not particular to this paper, by what is meant by tropical stability. It is a statement relative to temperate regions, which were covered by mile thick ice sheets. So the tropics were more stable than the temperate region. This does not mean that the tropical landscape and climate did not change dramatically. I am not sure if this is part of the basis for using the word stable in this paper.*

Response: Thanks for this clarification. We got the point and avoided the use of “environmental stability” in a general sense or without specification. The main reason for discussing landscape stability in this manuscript is underlining how global climate (expressed in precipitation changes across Amazonia) and sea level changes can cause important variations in the distribution of the distinct Amazonian riverine habitats. This goes against a general narrative that Amazonian drainage establishment dates to the Miocene, without major changes afterwards, and was driven by Andean uplift, often equating the origin of the transcontinental Amazon river to the origin of current Amazonian landscape configuration (see for example Hoorn et al. 2010, Albert et al. 2018, 2021). The position of major river valleys can be more stable in the Brazilian and Guiana shields, but there are evidences of great drainage rearrangement in the sedimentary lowlands (westward the Negro-Solimões confluence) during the Quaternary. So, the “stability” here refers to the view pointing to stable river courses after the establishment of the transcontinental Amazon River during the Miocene (or Pliocene for some authors). We have also made changes to the text to avoid misunderstanding (lines 57-60 and lines 270-336).

R#1-BSmith: *Line 59. “However, both views are limited by the lack of chronological information (i) to constrain the presumed environmental changes that split populations or (ii) to support landscape stability during the diversification of specific taxa.” I agree chronological information is useful, but it could also be really misleading if the population splits do not have anything to do with particular environmental changes.*

Response: Indeed, synchronicity does not necessarily mean cause-effect relationship. Here we are explicitly testing a temporal correlation coupled with habitat preference and

availability. Population increase at the same time of the expansion of its specific habitat (floodplains or archipelagos) would support a biotic response to landscape change. Temporal decoupling between demographic and landscape changes would reject the influence of the landscape change on the studied populations. Also, we are studying nine species, where five of them are associated with a very unique environment. Idiosyncratic events of demographic changes in each species, if unrelated to landscape change, would not result in consistent temporal distinction between the two ecological groups, nor in shared expansion within the group of island-specialists. We also aim to shed light on how the specific riverine environments respond to global changes (climate and sea level). In this way, we are able to establish a connection between global changes and population of species specialists in specific (and more restricted) environments. The discussion section about how birds responded to habitat availability (lines 270-336) was greatly revised to clarify this view.

R#1-BSmith: *And same as before, ref 9 does not say the landscape was stable. The word stable is not used in that manuscript (I just checked!). The study found that diversification was random with respect to landscape changes or occurred after the formation of the rivers and Andean uplift. The distinction is important.*

Response: Indeed, this is an indirect interpretation derived from the following sentence of the abstract on Smith et al. (2014):

"These results, augmented by the observation that most species-level diversity originated after episodes of major Andean uplift in the Neogene period, suggest that dispersal and differentiation on a matrix previously shaped by large-scale landscape events was a major driver of avian speciation in lowland Neotropical rainforests."

We interpreted that "a matrix previously shaped" imply stable landscape matrix during the period studied in that reference (Pleistocene), which encompasses the time frame studied in our manuscript. However, Smith et al. (2014) consider the Andean uplift as a driver of major features of the current landscape matrix, such as the valleys of large rivers and spatial distribution of uplands. Here, we are dealing with expansion and retraction of seasonally flooded environments within river valleys, which were highly dynamic during the Pleistocene and Holocene. We understand that the uplift of Andes is commonly evoked in the literature as the major control on the lowlands landscape because of its role to assemble the transcontinental Amazon River. Nevertheless, landscape changes related with the Andes uplift have a millions of years pace, which is beyond the thousand years changes of riverine environments reconstructed in this study. So, the Andes uplift was crucial to define the framework of the Amazon lowlands through the assembly of a whitewater transcontinental drainage, but the spatial distribution of specific habitats (floodplains, archipelagos and uplands on paleofloodplains) through time has other drivers performing on shorter timescales (thousand years timespan) leading to a more dynamic view about the landscape. This was the rationale we used to argue that we are describing new information that supports lack of stability. We understand this new geologic data presented here may change the paradigm for interpreting biological diversification in Amazonia. We have changed the text (introduction and discussion section about birds and their specific habitats - lines 270-336) to exclude general terms such as "stability" and avoid direct comparison between the landscape changes occurring in contrasting timescales (Andean uplift versus sedimentary budget within river valleys). This fast changing interpretation of Amazonian landscape

history is due to the lack of data about landscape processes along the Amazonian lowlands, a knowledge gap that is being partially reduced by the current study.

R#1-BSmith: *Line 146. It says UCE loci were used but in the Methods it says UCEs and two mtDNA were used. Were different sets of analyses done?*

Response: This was clarified in main text (lines 180-189) and methods section about bird genomics and demography (lines 458-551).

R#1-BSmith: *Discussion. Without tabs to start new paragraphs, it is harder to follow the start of new paragraphs.*

Response: We apologize for this. Tabs on paragraphs were added.

R#1-BSmith: *Line 177. What is meant by an opposite population dynamic is expected for seasonally flooded and upland forest species? Their population would have contracted? The demographic analyses I have seen from Lais Coelho's published dissertation shows concordant demographic expansion in upland forest birds.*

Response: We agree with the reviewer that "opposite" was not the most accurate term. We meant that distinct outcomes may be related to the same landscape process depending on how each species uses the habitat (lines 177-199). We have corrected the text and added information on demographic patterns of upland terra firme species (lines 274-282).

R#1-BSmith: *Line 261. "suggest recent contact of closely related populations in central Amazonia"*

Do you mean secondary contact and introgression? The sentence is unclear.

Response: Yes, it means introgression. The text was corrected (lines 309-312).

R#1-BSmith: *Line 263. "This evidence, combined with our demographic results, points to a dynamic history of changing connectivity of seasonally flooded environments in central Amazonia, suggesting that the current connectivity may have been recently attained." Isn't the expectation that river island habitats are ephemeral and that the rivers shift their courses. This is observable on Google Earth.*

Response: Indeed, islands are formed by large sediment bars that can grow or be partially eroded in a few years as observed in satellite images (but only bar tops are observed in satellite images) or appear and disappear from tens to thousands of years as pointed by our geochronological dataset. In the short timescale (years), the erosion of a given sediment bar (or part of the floodplain) can supply sediment for growth of another sediment bar downstream. This is the ubiquitous autogenic fluvial morphodynamics whose combination with controls action on longer timespans (sediment budget and channel width, for example) defines the spatial distribution of islands within a river channel. On the other hand, marginal floodplains are more dependent on drivers performing on longer timescale (thousand years and beyond), which define the amount of sediment trapped or eroded within river valleys. We show the effect of these different dynamisms (islands versus floodplains) on the history of specialized bird populations, i.e. we show that the dynamism affected floodplains connectivity through time causing differential demographic change. The alternative would be, for example, that populations remained connected even with the more dynamic habitat

changes (islands) with no demographic consequences, or that all floodplain species responded in the same way.

R#1-BSmith: Line 388. *“Afterward, each sample’s reads were mapped against reference alignments and phased (67).”*

Include more details. This explanation of how the data was phased is not repeatable. The same could be said for the entire bioinformatic sections. It is too brief to be repeatable. Is there a more detailed explanation in supplementary material?

Response: A more detailed description of bioinformatic procedures was included in the supplementary material (Bioinformatic procedures - File: SuppMaterial_Bioinformatics).

R#1-BSmith: Line 394. *If you have the entire mt genomes why use just ND2 and cyt b? There are decent references available.*

Response: We chose to use these gene regions because the evolutionary rates for them have been much more discussed and measured in the literature and thus they serve as good standards for comparison.

R#1-BSmith: Line 399. *In addition to checking for population structure, a test for isolation-by-distance is also warranted because IBD could also bias the demographic modeling. For example, species with IBD may show false positives of population expansion.*

Response: We included a test for IBD estimating the Effective Migration Surface (EEMS) of each species. All species show multiple areas of lower and higher migration than expected by IBD. Nonetheless, both sNMF and k-means classification suggest a better treatment of these species as single taxon. Therefore, the demographic signal presented here is not an artifact of either IBD or population structure. Please, see revised section "Demography of bird populations" (lines 177-199) and revised methods on bird genomic data (lines 458-551).

R#1-BSmith: Line 412. *What is the justification of requiring at least 4 informative sites? In order to avoid ascertainment bias, loci should be randomly selected. Very different demographic signatures can be inferred depending how loci subsampled. For example, preferentially selecting more variable loci is likely to exclude loci that do not show an expansion.*

Response: There aren't many published EBSM analyses with genomic data and by the time we were defining the analyses we used as reference results and tutorial from Trucchi et al. (2014). In this work, they compared the performance of EBSM using loci (RAD seq) with different degrees of polymorphism and show that “highly consistent results were obtained using separate datasets with 2, 3 or 4–6-SNP loci” as they state in page 3 but decided to use loci with 4-6 SNPs because of their higher information content. They also used one mtDNA (control region) with 50 nuclear loci to help in calibration. Because of the extensive tests they performed, we followed their approach. Furthermore, the additional analyses performed during the review process (see methods in lines 458-551) show that the pattern is maintained even with different analytical strategies.

Trucchi, Emiliano, et al. King penguin demography since the last glaciation inferred from genome-wide data. *Proceedings of the Royal Society B: Biological Sciences* 281.1787 (2014): 20140528.

R#1-BSmith: *The other issue with subsampling, which I assume was done because BEAST could not handle the size of your data, is that if it is only done once, it is unclear how much variance there would be across different subsamples. It is important to show a particular pattern is robust to a sampling procedure.*

Response: Yes, subsampling was done to make the analyses feasible and, although we performed several runs for each species to ensure convergence, it can only be done with the same input data. However, tests with different randomly selected UCEs recovered essentially the same results. Also, individual bias from subsampling loci in each taxon would hardly create a convergent pattern of demographic history across several taxa that matches distinct habitat affinities. Finally, ~2000 UCEs loci are also a small subsample of the genome. More importantly, the additional demographic analyses included during the review process show similar results of well supported demographic expansion, with concerted younger ages for island specialists. DILS method relies on estimating the summary statistics based on 1000 UCEs randomly selected and Ecoevolity uses all UCE loci. Please, see revised methods in lines 458-551.

R#1-BSmith: *The images and figures are fantastic. It might help to include a silhouette of one of the focal species on the skyline plots, so the genetic analysis stands out.*

Response: We agree that this would be nice, but there is already so much information in the figure that we decided to keep it as is.

R#1-BSmith: *Figures 4 and 5. The long tails of the right side of the skyline plot are not informative of the demographic history for the deeper (or lack thereof) coalescent events. The data has its limits. The legends do not explain to the reader how to interpret the skylines plots.*

Response: We decided to remove the skyline plots from Figure 4 (new Figure 7) and, as we agree with the reviewer, it was not very informative due to the time scale. We kept the whole plots in Figure 7 and improved the legend.

Brian T. Smith

American Museum of Natural History

Reviewer #2 (Remarks to the Author):

R#2: *I am an Ornithologist, and thus not qualified to comment on the geochemical analyses in this paper.*

That being said, I found the overarching story of the manuscript compelling, novel, and as a much-needed multidisciplinary perspective into biogeography; and an important contribution to our understanding of how populations interact with earth's history.

The detailed analysis of the geochemical history of the Amazon basin provides a novel context for understanding demographic history, and thus biography of amazonian birds. Biogeography has a long tradition of heuristic methods and inference, and so detailed and objective comparisons like this are consequential.

I believe this work does support the conclusions, although I have some comments about how the conclusions could have been more compelling (see below). The methods appear to be sound, from the biological perspective. I cannot comment on the geochemical analysis.

I only have a few general comments.

The first comment is that anthropogenic effects were not discussed. There is recent evidence that people may have drastically affected the Amazon up to ~10,000 years ago, which is within the timeline for expansion of some of these species, and human-mediated habitat change often results in edge and early-succession habitats that are favored by these species. I think this manuscript would be stronger if the authors consider the possibility of the role of anthropogenic effects, even if subjectively, or acknowledged this as a caveat.

Response: This is an interesting point. Thanks. There is ongoing debate about the extension and intensity of prehistoric human occupation on Amazonian forests. However, this debate is focused on upland forests. Following the line admitting that prehistoric humans significantly affected seasonally flooded forests through deforestation, occupation could favor expansion of species associated with early successional habitats. We do not have how to measure that, but we would expect more local expansion (not generalized expansion throughout the whole basin). We raised this hypothesis and have included this perspective as a possible alternative explanation (lines 283-290).

R#2: My second comment refers to biological sampling. I think that this study could have been strengthened by including a "control" group of upland avifauna. The paper mentions an explicit hypothesis that upland avifauna should show opposite patterns to island and floodplain avifauna, and the inclusion of these species would have the potential to provide a much more compelling story. As it is, many of the species show fairly similar demographic histories, and so the correlations may be conditionally independent on some other factor; for example a history of reticulate diversification and extirpation, independent of habitat and ecology. Separate species with similar distributions but that do not rely on these floodplain and island habitats showing drastically different demographic histories would greatly strengthen the inference in this analysis.

Response: We have added information on terra firme populations (lines 274-282) to the text (described in the reply above to Reviewer 1) and we have corrected the term "opposite" in the text, as we actually meant distinct effects. General comparisons between upland terra firme and floodplains taxa have been done before (Harvey et al. 2018) and in the present study we aim to refine this knowledge to understand the specific mechanisms that are leading to current patterns in the riverine environments (islands and floodplains), which are directly coupled to climate and sea level changes through fluvial sedimentation.

R#2: I also think a population phylogenetic approach could have helped this analysis, especially for island taxa. This could have provided diversification and demographic information within a larger context. For example, all island species show recent population expansions, and the authors mention this could have been due to increased connectivity of island areas. If these species were restricted to few and distant regions of islands, we would expect divergence times to coincide with this period, especially among species that show divergence within island taxa. Or, divergence from sister species not occupying islands may have occurred during previous periods of extensive river island habitat. Including within-

population diversification as well as timing of diversification from non-island sister species could provide much insight into the demographic history of these species.

Response: We agree and we have added information about sister groups and their distributions and habitat affinities to Table S2. We have also added this information to the discussion (325-336). Thank you again for the interesting suggestion.

Reviewer #3 (Remarks to the Author):

R#3: *This is a very interesting study that reports a series of dates from Amazonia in an attempt to assess the development of seasonally flooded habitats and the connection between flooded habitats and the evolution of endemic bird species. The study also assess the connection between regional climate, global sea level, the global carbon cycle and sedimentation in Amazonia.*

The results of this work are interesting and provide new and important insights into the connection between the sedimentary budgets of Amazonian lowland floodplains and the diversification of Amazonian bird species. The work also draws potential connections between the significant expansion of floodplain environments and a global increase in methane beginning at ~5 ka. Finally, the work connects sedimentary budget in Amazonia to both sea level and regional climate in a compelling way. The work has significant implications for understanding the development of the Amazonian basin, the diversification of its endemic taxa, and the connection between Amazonian sedimentary environments and the global carbon cycle, which in turns has potential implications for current and future climate change.

The work does a reasonably good job supporting its conclusions; however, the results could be laid out in a more compelling and convincing way with some revisions. In particular, the geochronology section would be improved if the following issues were addressed in revision:

R#3: *Introduction of the terrace environments and their dates earlier in the geochronology results section, rather than in the discussion of bird demographics. Although these dates are compiled from the literature, they sort of come out of nowhere when first introduced.*

Response: We appreciate your interest and comments about our manuscript. We changed the results and discussion sections to solve this inconsistency. Information about terraces and their literature ages are now presented in the results section (lines 151-161). A data table with upland terraces ages compiled from the literature was also added as supplementary material (Table S2).

R#3: *The explanation of rates of deposition, and in particular about "faster island formation" is not shown in a compelling way, partially due to the way the figures are illustrated and partially because not all of the information is really presented. I suggest that you could present this in a compelling way by showing some sediment thickness vs. dates in a figure and/or in a table showing sediment accumulation rates (and uncertainties).*

Response: We agree that sedimentation through time deserved a more robust presentation. So, we calculated sediment accumulation rates from vertical profiles dated in the Negro, Xingu and Solimões rivers. The depth-age profiles used to calculate accumulation rates

(aggradation) are now presented in Figure S3. Sedimentation rates and their uncertainties are shown in Figure S4 and Table S3. Figures S5 to S9 also show ages in vertical profiles distributed across the archipelagos in the Negro and Xingu rivers. The calculated sedimentation rates decrease from 2.38 to 0.33-0.01 cm/yr as the time interval increases from tens to hundreds-thousands years. This is in line with the unsteady character of sedimentation (Sadler 1981) and shows the large variation range of sediment accumulation, which results from decadal-secular episodic sediment inputs (short term) counterbalanced by progressive recurrence of erosive events in the millennial timescale (long term). However, the long term (millennial) expansion or retraction of floodplains and archipelagos is controlled by the slower rate and regional variation of base level (discharge + sea level). Thus, local changes resulting from episodic sedimentation have minor effects on the long-term (through the late Pleistocene and Holocene) landscape with spatial scale wide enough to isolate or establish contact of different bird populations. The millennial aggradation rates calculated for the seasonally flooded substrates (0.33-0.01 cm/yr = 3.3-0.1 m/ka) are in the range reported by the Sadler (1981) for terrigenous depositional systems. This sedimentation dynamics combining short-term sedimentation pulses and long-term long-term aggradation modulated by regional base level is now better described in the text (lines 162-173) and supported by additional data analysis presented in the supplementary material (Figures S3 and S4 and Table S3). In the revised version of the manuscript, we focused on the retraction and expansion of seasonally flooded habitats (floodplains and archipelagos) in the longer timescale (Holocene and late Pleistocene) and excluded interpretations about the abrupt millennial precipitation events as drivers of specific pulses of archipelago growth. We agree that interpretation of events occurring in 1-2 ka is limited by uncertainties of the geochronological datasets (age uncertainties and sampling/sedimentation rate bias).

Sadler, P. 1981. Sediment accumulation rates and the completeness of stratigraphic sections. *The Journal of Geology* 89(5), 569-584.

R#3: The stratigraphic figures are important, however, in their current form, Figures S5 and S6 are not very useful because there is no way to assess elevation/topographic/stratigraphic similarities or differences between the sections. The sections need to be shown relative to a datum to compare the sections and the dates.

Response: The spatially heterogeneous and diachronic character of sediment deposition under a period of relatively stable base level hinders chronostratigraphic correlations in the current (Holocene) seasonally flooded environments, including the archipelagos in the Negro and Xingu rivers. So, the main purpose of vertical sections presented in the supplementary material (now Figures S5 to S9) is to illustrate sedimentary facies forming flooded substrates and the time interval needed to stabilize sediment bars (demonstrated by muddy facies overlaying sandy facies).

We prepared a new figure (Figure 5) displaying the stratigraphic framework of the study area, which is represented by Holocene deposits of the current seasonally flooded areas overlaying older fluvial deposits forming upland terraces (lines 153-155). The upland deposits crop out in erosive riverbanks and were deposited from late Pleistocene in western Amazonia to Cretaceous-Miocene in central and eastern Amazonia. The Holocene deposits (substrates of seasonally flooded habitats) are unconformably covering the older fluvial deposits representing higher elevation incised terrains. The age gap between the Holocene and older upland fluvial deposits depends on the suitability for long-term preservation of

deposits accumulated within the Solimões-Amazon River valley. Preservation is reduced in central and eastern Amazonia because the Amazon River is incised in older Cretaceous-Miocene deposits showing higher elevation and higher resistance to erosion (Bezerra et al. 2021). This is the crucial stratigraphic aspect supporting the model to explain the expansion and retreat of seasonally flooded areas due to sediment input and base level changes across lowland Amazonia.

I.S.A.A. Bezerra, A.C.R. Nogueira, M.B. Motta, A.O. Sawakuchi, T.D. Mineli, A.Q. Silva, A.G. Silva Jr., F.H.G. Domingos, G.A.T. Mata, F.J. Lima, S.R.L. Riker. 2022. Incision and aggradation phases of the Amazon River in central-eastern Amazonia during the late Neogene and Quaternary. *Geomorphology* 399, 108073.

R#3: *It would also be useful to see some photos of what the deposits and the facies look like in situ.*

Response: Thanks for this observation. Indeed, the presentation of photos of the dated deposits is helpful to clarify text descriptions. We selected outcrop photos representative of the studied deposits and presented more detailed facies descriptions and interpretations in a new Results subsection named “Sedimentology of seasonally flooded substrates” (lines 114-146). Photos of the studied deposits are shown in new Figures 3 and 4.

R#3: *Figure 4, which is where the geochronology data is shown could be improved for clarity and to better emphasize the results of the work in several ways (1) panel 4B seems important, but I'm uncertain what its showing. Is this density of floodplain islands? If so, how was this determined? (2) In panel 4A, it is difficult to resolve different symbols and colors. Why two colors for each? (3) in panel 4A, what is the spread of dates on the y-axis indicating? (4) in panel 4F, add the location of each cave (i.e., eastern, northern, etc) to help clarify part of the discussion section.*

Response: We preferred to exclude panel 4B (now Figure 6) because sampling and preservation bias could influence probability density distribution of ages. So, new information was added (Figure 5 and Figures S3 and S4) to improve the stratigraphic representation of the geochronology data. Text was also updated to clarify the sedimentation dynamics responsible for the expansion and retreat of seasonally flooded areas (lines 148-173 and lines 239-266). Caption explaining symbols (OSL versus radiocarbon ages and this study versus literature ages) in panel 4A was added. The spread on the y-axis is an arbitrary offset to allow data visualization. This is now informed in the figure caption. Location of each cave was added in Figure S1 (previous Figure 1). Information on this was also added to the figure caption.

R#3: *Figure 5 would benefit from the addition of a sea level curve if possible.*

Response: This is now Figure 7. Indeed, the sea level curve in this figure is helpful to illustrate the discussion. It was added (panel 7D).

R#3: *Most of the discussion section is well reasoned, but I was wondering about the comment about episodic sedimentation increasing at 4 ka on line 209, because in the Tabuliero do Embaubal section it looks in some sections at 4 ka and then some between 2 and 3 ka?*

Response: Despite the island ages suggesting that the ria head archipelagos grew through sediment pulses (instead of constant rate progradation), defining the specific time periods of episodic sedimentation is difficult due to the sedimentation rate dependence on time span (lines 162-173). Reliable assessment of sedimentation rates through time would depend on the acquisition of sediment cores in sectors of the ria dominated by fine grained deposition and less susceptible to erosive processes (e.g. Bertassoli Jr. et al., 2019). Considering that the crucial issue is the expansion and retreat of flooded environments in the last glacial and deglacial periods (late Pleistocene and Holocene), the text was changed (lines 239-253) in order to highlight the archipelago growth since the mid Holocene and discuss sedimentation rate in terms of the time span (centennial to millennial).

Bertassoli Jr, D.J., Sawakuchi, A.O., Chiessi, C.M., Schefuß, E., Hartmann, G.A., Häggi, C., Cruz, F.W., Zabel, M., McGlue, M.M., Pupim, F.N. 2019. Spatiotemporal variations of riverine discharge within the Amazon Basin during the late Holocene coincide with extratropical temperature anomalies. *Geophysical Research Letters* 46, 9013-9022.

R#3: Finally, in the discussion section, the potential connection between methane concentrations and flood plains is a really interesting idea. I encourage the authors to develop this idea a bit more because it would definitely increase the significance of the work.

Response: Thanks for this suggestion. We improved the discussion about floodplains extension and methane emissions. The new version of the manuscript now has a section entitled “Floodplains extent and atmospheric methane concentration” (lines 340-358). However, we prefer to not go deeper in this issue to keep a focus on the relation between fluvial sedimentation and bird demography. However, we hope to revisit the relation between Amazon floodplains and past methane emissions in a new future manuscript.

R#3: The methodology is sound and reasonable presented. The methods section would benefit a bit from some discussion about how the vertical profiles were collected (i.e., cores, sediment pits, something else?).

Response: Samples were collected in outcrops, pits or using a manual auger adapted to use plastic liners. More detailed information on sediment sampling was added to the geochronology methods section (lines 401-410).

Reviewer #4 (Remarks to the Author):

R#4: As a geologist that works on rivers and their evolution, I will confine my comments to the geological part of the paper, rather than comment on the more biological aspects. The latter are very interesting, but I know nothing about them.

With that said, as a geologist that works on rivers, I have a hard time assessing what the authors have done. Granted, this is a tough environment to work in, but I think a lot of the fluvial part of their story is overinterpreted relative to documentation of their story. I have no issue with the coupling of, or separation of, the effects of sea-level change and climate on river behavior, both types of drivers are inherently related over the time scales of their study.

Response: Indeed, rivers are very large and sampling sites are hard to reach, needing many and costly field surveys. This manuscript integrates data acquired during several field surveys in the Solimões-Amazon main stem and its tributaries since 2011, besides data from the literature. So, this is a huge group effort. We agree that differentiating the effects of sea level and precipitation changes on river behavior is a hard task when we go over millennial timescales (late Pleistocene and Holocene), which is the main purpose of our work (besides the effect of this on bird populations). Climate (precipitation) and autogenic fluvial processes drive decadal to secular sediment accumulation and preservation. This is now better demonstrated (lines 162-173 and lines 209-213) by age-depth profiles and sedimentation rates presented in new Figures S3 and S4. The main purpose of our work regarding river behavior is to understand the mechanisms leading to expansion and retreat of seasonally flooded substrates, which is driven by sediment accumulation or erosion within river valleys. We observed marked elevation and age differences between deposits of the seasonally flooded substrates and the adjacent terraces representing ancient floodplains. This is now better illustrated in new figures (Figure 1 and 5). The upland fluvial deposits have deposition ages older than ~45 ka in western Amazonia (Pupim et al., 2019), but can be much older (late Cretaceous-Miocene) in central-eastern Amazonia (Bezerra et al. 2021) while the floodplains within the river valleys are Holocene (< 11 ka) both in western and central-eastern Amazonia (see Figures 5, 6 and 7). These results point to a base level fall (incision) since ~45 ka (youngest age recorded in fluvial terraces bounding the Solimões River in western Amazonia) and base level rise (aggradation) during the Holocene (< 11 ka). The critical issue is to differentiate between the roles of sea level and precipitation (river discharge) changes for these last incision (45-11 ka) and aggradation (< 11 ka) phases. Abrupt millennial precipitation events could be important to input sediments to river valleys (in the bar/island scale), but they are unsuitable to promote sustained erosion or aggradation over the time frame of tens of thousands years. For lowland Amazonia, we interpret that reduced sediment budget within river valleys (incision) occurred from 45 ka to the LGM when sea level was falling and precipitation was high while aggradation (positive sediment budget) occurred by sea level rising/highstand combined with higher precipitation, condition prevailing during the Holocene (Figures 6 and 7). Hence, the expansion and retraction of floodplains across lowland Amazonia would be a combined effect of sea level and climate. In the new version of the manuscript, we tried to clarify this view in the discussion section (lines 205-266) and recognize limitations imposed by the completeness of the sedimentary record, which hampers comparisons with short term (1-2 ka) precipitation events, such as abrupt millennial shifts.

R#4: *But I have a difficult time evaluating the stratigraphic context and significance of their geochron data, and the reasonableness of their interpretations about rates and scales of fluvial processes over time. I therefore cannot fully evaluate their story, and therefore cannot recommend the paper for publication without major revisions.*

Response: We calculated sedimentation rates from age-depth profiles and prepared a new figure (Figure 5) to better illustrate the stratigraphic context of the target deposits (floodplains and their bounding upland terraces). Density plots of sediment deposition ages were excluded from Figure 6 (previous Figure 4) because they can be biased toward higher preservation of younger deposits, as demonstrated by sedimentation rates (Figures S3 and S4 and Table S3). Sedimentation rates demonstrate a sedimentary dynamics characterized by short time (decadal-centennial) pulses of local sediment accumulation (bar/island scale),

which are reworked by further erosive processes, conditioning preservation to long term (millennial) base level changes. Thus, base level drives aggradation and incision in the regional (millennial) scale. Figure 5 shows the basic stratigraphic context supporting this view, which is characterized by Holocene fluvial deposits within river valleys incised in upland fluvial terraces representing past floodplains. Please, see new results section about chronology of sediment deposition (lines 148-173) and the discussion section about expansion and retraction of seasonally flooded environments (lines 205-266).

R#4: *Here are a few specific comments:*

121-135 – I am uncomfortable with inferences about rates from a data set like this. This paragraph infers things like “highly intensified”, a “retreat phase”, “faster island formation”, and “.....show depositional and colonization rates varying.....”. I am not really clear how they arrive at these rate-based interpretations, other than the sheer numbers of ages for specific time periods. The enormous and impressive number of OSL and 14C ages is, in itself, an important dataset, but it’s hard to see the context within which they can be interpreted in terms of rate-like statements.

Response: The whole text was revised in order to eliminate subjective terms or statements not supported by the presented dataset. Sedimentation rates were (Figures S3 and S4 and Table S3) calculated to support the discussion about the sedimentary dynamics over decadal-centennial versus millennial scales (lines 162-173). The intent of the ages plots (Figures 6A and 7A) was to show the age gap (incision phase) between the current seasonally flooded deposits and higher elevation fluvial terraces. We agree that age distributions are influenced by sampling, but ages in Figure 7A are group in terms of their parent substrate (floodplains or terraces) and a new figure (Figure 5) was prepared to illustrate the stratigraphic context of the age gap between Holocene seasonally flooded substrates and their bounding Pleistocene or older terraces corresponding to paleofloodplains. This supports the variations in the extension of floodplains during the late Pleistocene and Holocene. This is the critical aspect for comparison with bird populations demography. Other issues about episodic sedimentation in shorter timescales were reduced and clarified in the results (lines 162-173) and discussion (lines 205-266) sections.

R#4: *179-194 - It is almost a certainty that the entire valley and the channel were at a significantly lower elevation for much of the glacial-period sea-level lowstand, even this far upstream, and especially during the LGM. An interesting paper by Mertes and Dunne (I think), speculates on the lowstand long profile of the Amazon, which is consistent with the well-defined lowstand long profile of the Mississippi. The incision tapers upstream, of course (as the authors recognize), but it can extend a long ways, and I suspect, as the authors discuss, this area is still in the part of the system that aggraded a lot during the latest Pleistocene and early Holocene period of rapid sea-level rise, such that any records of flood plains from that time would be buried, or at the very least, the record of those times is biased against preservation and discovery using the methods they use. Moreover, most rivers have several terraces that represent that period of incision, such that the incised valley topography has lots of variability, the glacial period flood plain would have been inset into and confined by those terraces. The broad flood plain of the late Holocene developed over the top of, and therefore obscures, that variability because it is now only confined by higher valley walls.*

Response: We agree. The supporting information and discussion about the roles of sea level changes and climate for formation and preservation of fluvial deposits representing seasonally flooded lands and uplands were improved in the new version of the manuscript (lines 205-266). In fact, LGM lower elevation terraces could be buried by Holocene deposits accumulated during sea level rising (and precipitation increase), especially in the area of the middle Solimões river in western Amazonia, where the wider valley increases preservation of older deposits. In this area, Pleistocene floodplains are represented by the Içá Formation, with a minimum age of ~45 ka (Pupim et al., 2019). So, eventually preserved low elevation LGM deposits should be buried by the Holocene floodplains sediments (see profiles in new Figure 5). The preservation potential of older fluvial deposits is reduced in central and eastern Amazonia (downstream the Solimões-Negro river confluence) because river valleys are narrower and incised in Late Cretaceous-Middle Pleistocene deposits more resistant to erosion (Bezerra et al. 2021). The hypothesis of a lower elevation base level for the central-eastern Amazonia at the LGM proposed by Irion et al. (2009) is highly speculative and based on very little observational support. Their main argument is the presence of sediment-buried paleo-valley surfaces in ria lakes (the drowned valleys of some tributaries to the Amazon). Since there are no ages for such valley infills (only ages for the upper 5-10 m infill are available – see depth-age profiles for the Negro and Xingu rias and ages presented for the Xingu ria in Bertassoli Jr. et al. 2017, 2019), the interpretation of past profiles for the Amazon through time is clearly model-driven. On the contrary, high-resolution Multibeam-Ecosounder surveys (Almeida et al. 2016, Galeazzi et al. 2018, Gualtieri et al. 2020), and river bottom sampling reveal that the talwegs in single-channel reaches in the lowermost Solimões River, the Solimões-Amazon junction, and the upper Amazon are actually exposing the bedrock (Cretaceous sandstone). Therefore, a supposed aggradational phase of the whole valley since the LGM is not possible in the target area. This evidence is in agreement with the here discussed chronological evolution of river bars and floodplains.

Almeida, R.P.; Galeazzi, C.P.; Freitas, B.T.; Janikian, L.; Ianniruberto, M.; Marconato, A. 2016. Large barchanoid dunes in the Amazon River and the rock record: Implications for interpreting large river systems. *Earth and Planetary Science Letters* 454, 92-102.

Bertassoli, D.J.; Sawakuchi, A.O.; Chiessi, C.M.; Schefuß, E.; Hartmann, G.A.; Häggi, C.; Cruz, F.W.; Zabel, M.; McGlue, M.M.; Santos, R.A.; Pupim, F.N. 2019. Spatiotemporal Variations of Riverine Discharge Within the Amazon Basin During the Late Holocene Coincide With Extratropical Temperature Anomalies. *Geophysical Research Letters* 46, 9013-9022.

Bertassoli, D.J.; Sawakuchi, A.O.; Sawakuchi, H.O.; Pupim, F.N.; Hartmann, G.A.; McGlue, M.M.; Chiessi, C.M.; Zabel, M.; Schefuß, E.; Pereira, T.S.; Santos, R.A.; Faustino, S.B.; Oliveira, P.E.; Bicudo, D.C. 2017. The Fate of Carbon in Sediments of the Xingu and Tapajós Clearwater Rivers, Eastern Amazon. *Frontiers in Marine Science* 4, 44.

Bezerra, I.S.A.A., A.C.R. Nogueira, M.B. Motta, A.O. Sawakuchi, T.D. Mineli, A.Q. Silva, A.G. Silva Jr., F.H.G. Domingos, G.A.T. Mata, F.J. Lima, S.R.L. Riker. 2022. Incision and aggradation phases of the Amazon River in central-eastern Amazonia during the late Neogene and Quaternary. *Geomorphology* 399, 108073.

Galeazzi, C.P.; Almeida, R.P.; Mazoca, C.E.M.; Best, J.L.; Freitas, B.T.; Ianniruberto, M.; Cisneros, J.; Tamura, L.N. 2018. The significance of superimposed dunes in the Amazon River: Implications for how large rivers are identified in the rock record. *Sedimentology* 1, 1.

Gualtieri, C.; Martone, I.; Filizola, N.; Ianniruberto, M. 2020. Bedform Morphology in the area of the confluence of the Negro and Solimões-Amazon Rivers, Brazil. *Water* 12, 1630.

Irion, G., Müller, J., Morais, J.O., Keim, G., Mello, J.N., Junk, W.J. 2009. The impact of Quaternary sea level changes on the evolution of the Amazonian lowland. *Hydrological Processes* 23, 3168-3172.

R#4: 206-210 – *I do not see how such interpretations come from this type of dataset. What if they were able to date every layer, would things still be interpreted as episodic? Over what time scale does something become episodic - decades, centuries, millennia? And when one bar was not active, did another bar somewhere else become more active?*

Response: Interpretations and discussion about episodic sedimentation was clarified with a constrained timescale. “Episodic” was used in the sense of abrupt or low frequency/high intensity event, analogous to abrupt millennial precipitation events (Heinrich Stadials, for example) compared to precession precipitation variations. This approach was adopted assuming that abrupt precipitation events promote episodic input of sediments to river valleys. However, deposits recording episodic sediment input are further reworked by regular autogenic processes if base level stays stable over the timescale. This is the case of the studied system, as demonstrated by the increase in sedimentation rates as the time span decreases (Figures S3 and S4 and Table S3). The decadal-centennial time scale represents processes driving bar growing and stabilization while the regional expansion or retraction of floodplains occur in the millennial timescale driven by base level changes (sea level and precession water discharge). The text was revised (lines 148-173 and lines 205-266) to clarify the sedimentary dynamics over these two (decadal-centennial and millennial) timescales.

R#4: 295-300 – *Absolutely this dataset is biased by differential preservation, and the ensemble of ages they cite do reflect a bias towards younger ages - that's the normal case. One thing that would be useful in this respect would be to actually map the terraces and bars that were active at different times, and therefore have weaker or stronger development of forest – maybe by millennia or something, or even plot samples locations by age – different colors for each millennium or something. One can then evaluate the spatial scales of preservation. But right now, I see no way to evaluate changes through time because the geochron is not really tied to an independently developed stratigraphic model. Instead, they have a large number of ages, but it's difficult to see what they actually mean. In any migrating river, preservation of a bar and overbank fines is tough due to constant migration and reworking. The surprise would be if something else would have emerged, not that this common bias towards the preservation of younger strata is there.*

Response: Indeed, calculated sedimentation rates (Figure S4) demonstrate the preservation bias toward younger deposits (higher sedimentation rates). However, the main purpose of the geochronological data was to characterize the ages of the seasonally flooded substrates currently occupied by igapó and várzea forests as well as older seasonally flooded substrates (paleofloodplains), which are now upland terraces. Figure 2 was changed to show the difference in elevation between floodplains within valleys and the neighboring terraces. A new figure (Figure 5) was also prepared to present stratigraphic-relief profiles showing the age difference between the seasonally flooded sediments and the neighboring terraces. We believe that the ages presented in Figures 6A and 7A are now supported by a stratigraphic and geomorphologic context.

R#4: *Figures 1-3 are low res DEMs and satellite images that are not really that informative but take a lot of space that might otherwise be used for data or analyses. Moreover, generally, the figure captions need more detail.*

Response: The purpose of Figure 1 is to give a regional overview of the study areas and to present sample and precipitation record locations. We agree that it is not adding relevant information to results or discussion. So, it was moved to the supplementary material (Figure S1). A new Figure 1 was prepared to better illustrate the surface morphology of floodplains and neighboring terraces as well as to give context to the stratigraphic profiles shown in Figure 5. Figure captions were revised to include more details.

R#4: *Figure 4 – 4A is not an effective way to plot these data. I am not sure what 4B is, from the caption I was expecting radiocarbon ages. But they are included in 4A. And the Y-axis needs to have units, I have no idea with regards to B what density refers to?*

Response: The intention of Figure 4A (now Figure 6A) is to allow comparison with other variables plotted through time (sea level, precipitation, demography etc.). We excluded the probability density curves because of the sampling and preservation bias (as discussed in the sedimentation rates issues in lines 1620173). The plots of sedimentation ages through time (Figures 6A and 7A) are now supported by new spatial and stratigraphic data representation in Figures 1 and 5, besides the supplementary Figures S5 to S9 (ria archipelagos).

R#4: *Figure 5 – I am not sure what to make of this figure, they are arguing for large-scale correlation and causality, but it's hard to understand this from the text, and the data, and it therefore seems like sort of an eyeball or qualitative trend matching test, not anything rigorous that shows cause and effect. This was the norm a while ago, but not so much now.*

Response: Statistical/time series analyses could be applied to check the correlation between climate proxies. This has been done intensively in previous works. Here, the challenge is the representation of fluvial landscape changes (recorded by sedimentation ages), climate/sea level changes and birds demography in the same timescale. These three types of information have different age uncertainties and completeness, which makes the application of statistical correlation analysis difficult, with advantages beyond a comparative graphical approach. This limitation is considered in the discussion section entitled "Limitations and future perspectives" (lines 362-396). Despite this limitation, the representation of regional landscape changes, climate/sea level and birds demography in the same timescale allows empirical evaluation on the hypothesized relationship between fluvial dynamics, precipitation/sea level and bird populations. Previous studies presented only general statements on this relationship, with very poor age constraints. Thus, we believe this new information together allows us to reach a new perspective about the crucial question of how specialized birds respond (or not) to long-term changes in their specific habitats.

REVIEWERS' COMMENTS

Reviewer #1 (Remarks to the Author):

I previously reviewed this manuscript, which is on the sedimentary history of floodplains in Amazonia, the chronology of the formation of seasonally flooded habitats, and population history of birds occurring in the habitats associated with these regions. The current draft is really improved. My previous recommendations were largely addressed in the point-by-point response and in the text. I think the addition of new demographic and spatial analyses make the paper more rigorous. The authors original conclusions holdup after more in depth modeling of the genomic data.

My only remaining concern is that there is limited discussion of the genetic results. Where appropriate in the Results and Discussion this could be expanded or maybe in Supplementary text. The reader is left to interpret, understand, and accept the genetic analyses, particularly the ones that were added during review by looking at supplementary figures. I suspect there are a lot of nuanced details from the new analyses and quality control steps that would be really informative to the reader.

In sum, I would like to add this is quite a novel study design and represents what future biogeographic studies may look like. For a long time researchers have discussed combining geological and genetic data, but this work represents a clear leap forward. A job well done.

Additional comments:

Line 195. A more in-depth discussion on the ecoevolity and DILS results are warranted. There is no mention of the specific results of Fig S11, which shows six expansion events. I agree the results largely support the scenario outlined of a late expansion in river island birds, but the nuanced details are worth mentioning. There is no evidence of synchronous expansion in the floodplain species and not all river-island birds likely expanded at the same time, although they all expanded recently.

Demography of bird populations Section. There are several undefined acronyms in this section that may confuse readers (e.g., DILS, UCEs, EBSP)

Line 379. "Particularly, estimating precise ages of population expansion events is still limited by the uncertainties regarding molecular evolutionary rates". and generation times can be added.

We do not have a good handle on generation times and the variation among species, which can have an impact on the degree of synchronicity and absolute expansion.

Line 382. phylogeographic inferences to gest geological hypothesis (71, 72). → typo. Change gest to test.

The challenges with testing geological hypotheses is much larger than estimating accurate substitution rates. But the topic is better off for another paper.

Line 469 "and trimmed to improve alignment quality." → Can you be more specific on what was done and how it was done?

Line 469 "Samples with low coverage were discarded" What was considered low-coverage?

Line 503. Include justification for including loci with 4 informative sites. I do not necessarily agree with the justification in the point-by-point response, but it is important that the rationale be explained to readers, especially for those that might use this paper as justification for their experimental design.

Paragraphs starting at lines 527 and 538. The text in these paragraphs can be clearer. There are few examples of awkward word choices or phrases.

Line 538. "For that, we used the complete phased UCE data, including constant sites, concatenated in a single nexus file per species. The authors recommend this approach as it improves Ecoevolity performance (35)." -> Simplify

For that, we used the complete phased UCE data, including constant sites, concatenated in a single nexus file per species, as recommended by the software's authors.

Line 544 "probabilities for most numbers of events," Clarify what is meant in this phrase. I know what you are referring to, but it can easily be misunderstood by readers.

Figure 7. Scientific names should be in italics

Fig S12. The legend should state which analysis (i.e., ecoevolity) the results are from.

Figure S13. Estimated Effective Migration Surface (EEMS). What is the reader supposed to take from these images? The plots are pretty and they show gene flow is not uniform across space, which is the expectation. But what can be inferred from the maps?

Brian T. Smith
American Museum of Natural History

Reviewer #2 (Remarks to the Author):

In my view, the authors have adequately addressed my and other reviewers' concerns, within the scope of the data and abilities of their project.

Reviewer #3 (Remarks to the Author):

The authors have done an excellent job revising this manuscript and have addressed all of my major comments. The new figures and the detailed descriptions of the facies and age model are an excellent addition to the paper and the supplement. The geological data presented here is compelling and supports the interpretations about connections between diversification of bird populations and the development of seasonally flooded habitats in Amazonian systems. The results are very interesting and compelling. The authors also did a nice job of connected larger global changes in sea level and regional changes in the SAMS to deposition in the Amazonian system. The connection to the global methane record is also interesting and is a good addition - though perhaps could be softened just a bit.

I only have a few minor comments listed below:

When discussion sediment accumulation of flood plains, you used the term "synchronic". It should be "synchronous".

Perhaps scale this comment back a bit: "likely had a central impact on global CH₄ budgets" to be something like "may have played an important role on global CH₄ budgets". The connection is certainly interesting and potentially there, but more work needs to be done to confidently connect the expansion of Amazonian flood plains to CH₄ budgets.

Check for typos throughout. For example, "stratigraphich" instead of "statigraphic" in the "Limitations..." section

Reviewer #4 (Remarks to the Author):

GENERAL COMMENTS

I reviewed the first version of this paper in the Summer of 2021. I thought it was an interesting paper that was inherently multidisciplinary, and would have impact beyond any single field. However, because of my expertise, I restricted my comments to the geological aspects. I restrict my comments here too for the same reason.

I also recognized that this was a tough environment to work in, but felt like the fluvial part of their story was overinterpreted RELATIVE to the data they presented. For this revision, the authors have made a concerted effort to address these issues. In short, they have greatly improved presentation of the data, especially documentation of the context of their geochronological data, and the types of environments and facies that they are working with. With that said, there are a couple issues that might warrant further clarification, especially as they relate to the influence of climate-forced "glacio-eustatic" sea-level change. At this point, I see this as issues of clarification, rather than issues that require a major revision. So I recommend publication, and think the paper would be further improved by one more minor set of revisions.

ON THE INFLUENCE OF SEA-LEVEL CHANGE

While their geological framework is much improved, and their interpretations have sharpened significantly, I think they still downplay the effects of river long profile response to sea-level change. For me, this is the fundamental physical reality that provides context for the many valuable observations they make. And in the for what its worth category, they cite the Blum and Tornqvist (2000) paper on this issue, but I suggest the authors might find the Blum et al., 2013 Earth Science Reviews paper to be more specific about such issues.

Regardless, both papers make the case that rivers have a predictable response to sea-level lowering - they incise their channels and valleys, as they extend their channels to lower sea-level and shoreline positions. This response tapers to zero upstream over a distance that scales to gradient and flow depth. Conversely, rivers have a predictable set of responses to sea-level rise - they must (a) aggrade and fill their valleys to keep pace with sea-level rise, or (b) form a drowned valley estuary if sediment supply is low. The distance over which this aggradation occurs also scales to gradient and flow depth - the minimum distance is roughly approximated by the backwater length which scales as $L_b \sim \text{mean water depth} / \text{slope}$. But, while the base of the channel responds upstream over that distance, the flood plain must also feel the effects. For example, if channel X is 25 m deep when the base of the channel descends below zero elevation, i.e. the backwater length, the flood plain height would be ~ 25 m elevation.

I agree with the authors that their Solimoes sites are pretty far upstream, but I would hypothesize that the river flood plain is still feeling the effects of base level rise. Coari, one of the places identified in the paper, is ~ 1850 river km from the mouth - I am not sure of the channel depth at that point, but the water surface in GoogleEarth is ~ 20 m elevation, so it is still likely within the reach that is affected by backwater hydraulics, and I would assume the aggradational drive from sea-level rise is still being felt there. Additionally, consider the gradient of the Solimoes is < 0.00005 from Coari to Tefe. This is approximately the same gradient the Amazon River downstream from Manuas must attain at sea-level lowstand to reach the shelf edge canyon at depths of ~ 100 m or so. My basic point is that it is plausible that flood plains throughout their entire study area feel the effects of climate-forced sea-level change.

For me, the incision and aggradation, which created the valley that has since been mostly filled with wetland environments like those discussed in the paper, is also part of the globally- and regionally-relevant climate story. I would argue this is the first-order explanation for the temporal distribution of their geochron data, including the minimal numbers of ^{14}C or OSL ages during isotope stage 4, 3 and 2, and the peak number of ages during the late Holocene, as shown in Figures 6 and 7. But I would argue that such millennial-scale expansion of wetlands likely occurred during isotope stages 4,3 and 2, its just that the record is now buried. They mention this bias, but the methods they use are fundamentally biased to document deposits that represent global sea-level highstand. This does not diminish the significance of what they document in any way, what they have done stands on its own merit. But what they have documented could also be of a

fundamental response of the river system's flood plain that likely occurred during all stages of sea level as the channel and flood plain adjust their profile to the contemporaneous sea-level and shoreline position.

I also have long thought it likely that the many flooded river valleys that join the main channels are themselves drowned valleys because of aggradation of the main river channels – I do not know the literature enough to know whether this is incorrect or common knowledge. But I suspect they cannot keep up with the Andean sediment loads present in the main channels, so do not fill their valleys as fast as the Amazon has, but their very presence is likely also a consequence of climate-forced sea-level rise and drowning of glacial-period valleys.

In summary, I think changes in climate of the type discussed in the paper are very important, interesting, and solidly interpreted – what they discuss merits publication in Nature COMMS. But I also think all of those effects are overlain on this broader contextual stage that represents climate-forced global sea-level change and its effects on the Amazon river system, and it would be nice to see that featured in the paper as well.

SPECIFIC COMMENTS

I think the graphics are greatly improved, and, within a difficult environment, now display a lot of the important context for the geochron data that underlies their interpretations.

I think they need a general location map for the main text. As it is now, the reader has to go to the Supplementary Material, Figure S1, to see the locations of study areas within the context of the river system. Figure 1 in the text just shows local context, which is very useful, but the reader will have to go to the Figure S1 to see where the locations are.

RESPONSES TO REVIEWERS' COMMENTS (OUR RESPONSES ARE IN ITALICS)

Reviewer #1 (Remarks to the Author):

R#1: I previously reviewed this manuscript, which is on the sedimentary history of floodplains in Amazonia, the chronology of the formation of seasonally flooded habitats, and population history of birds occurring in the habitats associated with these regions. The current draft is really improved. My previous recommendations were largely addressed in the point-by-point response and in the text. I think the addition of new demographic and spatial analyses make the paper more rigorous. The authors original conclusions holdup after more in depth modeling of the genomic data.

My only remaining concern is that there is limited discussion of the genetic results. Where appropriate in the Results and Discussion this could be expanded or maybe in Supplementary text. The reader is left to interpret, understand, and accept the genetic analyses, particularly the ones that were added during review by looking at supplementary figures. I suspect there are a lot of nuanced details from the new analyses and quality control steps that would be really informative to the reader.

Response: *More details about the genomic analyses were added in the last section of the Results ("Demography of bird populations") and where appropriate, as described below in response to the specific comments.*

R#1: In sum, I would like to add this is quite a novel study design and represents what future biogeographic studies may look like. For a long time researchers have discussed combining geological and genetic data, but this work represents a clear leap forward. A job well done.

Response: *We are glad the interdisciplinary work presented here was considered by the reviewer as a significant contribution to the field!*

R#1: Additional comments:

R#1: Line 195. A more in-depth discussion on the ecoevolution and DILS results are warranted. There is no mention of the specific results of Fig S11, which shows six expansion events. I agree the results largely support the scenario outlined of a late expansion in river island birds, but the nuanced details are worth mentioning. There is no evidence of synchronous expansion in the floodplain species and not all river-island birds likely expanded at the same time, although they all expanded recently.

Response: *We added a more thorough explanation of the results obtained in the new analyses, including more detailed interpretation of the results shown in Supplementary Figures 10, 11 and 12 (lines 195-216).*

R#1: Demography of bird populations Section. There are several undefined acronyms in this section that may confuse readers (e.g., DILS, UCEs, EBSP)

Response: *We reduced the use of acronyms to make reading less confusing.*

R#1: Line 379. "Particularly, estimating precise ages of population expansion events is still limited by the uncertainties regarding molecular evolutionary rates". and generation times can be added. We do not have a good handle on generation times and the variation among species, which can have an impact on the degree of synchronicity and absolute expansion.

Response: We agree and included generation time as a source of uncertainties in recovering accurate ages of population expansion.

R#1: Line 382. phylogeographic inferences to gest geological hypothesis (71, 72). → typo. Change gest to test.

Response: Corrected.

R#1:The challenges with testing geological hypotheses is much larger than estimating accurate substitution rates. But the topic is better off for another paper.

Response: We agree. Congruent ages between geological and demographic events do not necessarily indicate causality. However, more accurate estimates of the ages of these events provide better data to test hypotheses regarding the influence of dated geological events on biogeographic histories, and may reject causal links between events with disagreeing ages with larger confidence.

R#1: Line 469 “and trimmed to improve alignment quality.” → Can you be more specific on what was done and how it was done?

R#1: Line 469 “Samples with low coverage were discarded” What was considered low-coverage?

Response: A more detailed explanation of the genomic data processing was included in the Methods section (**Bird genomic data sampling and historical demographic analyses**).

R#1: Line 503. Include justification for including loci with 4 informative sites. I do not necessarily agree with the justification in the point-by-point response, but it is important that the rationale be explained to readers, especially for those that might use this paper as justification for their experimental design.

Response: A justification of this approach was included in this section of the methods to explain the rationale (lines 531-535).

R#1: Paragraphs starting at lines 527 and 538. The text in these paragraphs can be clearer. There are few examples of awkward word choices or phrases.

Line 538. “For that, we used the complete phased UCE data, including constant sites, concatenated in a single nexus file per species. The authors recommend this approach as it improves Ecoevolity performance (35).” -> Simplify

For that, we used the complete phased UCE data, including constant sites, concatenated in a single nexus file per species, as recommended by the software’s authors.

Line 544 “probabilities for most numbers of events,” Clarify what is meant in this phrase. I know what you are referring to, but it can easily be misunderstood by readers.

Response: Agreed. The aforementioned paragraphs have been adjusted to make reading more straightforward.

R#1: Figure 7. Scientific names should be in italics.

Response: Corrected.

R#1: Fig S12. The legend should state which analysis (i.e., ecoevolity) the results are from.

Response: The legend was adjusted to include this information.

R#1: Figure S13. Estimated Effective Migration Surface (EEMS). What is the reader supposed to take from these images? The plots are pretty and they show gene flow is not uniform across space, which is the expectation. But what can be inferred from the maps?

Response: We elaborate further on the EEMS analyses in the results section (lines 186-192).

Brian T. Smith

American Museum of Natural History

Reviewer #2 (Remarks to the Author):

R#2: In my view, the authors have adequately addressed my and other reviewers' concerns, within the scope of the data and abilities of their project.

Response: Thank you.

Reviewer #3 (Remarks to the Author):

R#3: The authors have done an excellent job revising this manuscript and have addressed all of my major comments. The new figures and the detailed descriptions of the facies and age model are an excellent addition to the paper and the supplement. The geological data presented here is compelling and supports the interpretations about connections between diversification of bird populations and the development of seasonally flooded habitats in Amazonian systems. The results are very interesting and compelling. The authors also did a nice job of connected larger global changes in sea level and regional changes in the SAMS to deposition in the Amazonian system. The connection to the global methane record is also interesting and is a good addition - though perhaps could be softened just a bit.

Response: We changed some sentences to have less emphasis on the methane issue, and also clarified in the "Floodplains extent and atmospheric methane concentration" section that specific studies are needed to move on this problem.

R#3: I only have a few minor comments listed below:

When discussion sediment accumulation of flood plains, you used the term "synchronic". It should be "synchronous".

Response: Done.

R#3: Perhaps scale this comment back a bit: "likely had a central impact on global CH₄ budgets" to be something like "may have played an important role on global CH₄ budgets". The connection is certainly interesting and potentially there, but more work needs to be done to confidently connect the expansion of Amazonian flood plains to CH₄ budgets.

Response: This issue was rephrased in a more cautious way as suggested. The following sentence was also added to the end of the "Floodplains extent and atmospheric methane concentration" section: "However, specific studies are needed to confidently test the connection between the expansion of Amazonian floodplains and global CH₄ budgets."

R#3: Check for typos throughout. For example, "stratigraphich" instead of "statigraphic" in the "Limitations..." section

Response: Typos were double checked.

Reviewer #4 (Remarks to the Author):

GENERAL COMMENTS

R#4: I reviewed the first version of this paper in the Summer of 2021. I thought it was an interesting paper that was inherently multidisciplinary, and would have impact beyond any single field. However, because of my expertise, I restricted my comments to the geological aspects. I restrict my comments here too for the same reason.

I also recognized that this was a tough environment to work in, but felt like the fluvial part of their story was overinterpreted RELATIVE to the data they presented. For this revision, the authors have made a concerted effort to address these issues. In short, they have greatly improved presentation of the data, especially documentation of the context of their geochronological data, and the types of environments and facies that they are working with. With that said, there are a couple issues that might warrant further clarification, especially as they relate to the influence of climate-forced “glacio-eustatic” sea-level change. At this point, I see this as issues of clarification, rather than issues that require a major revision. So I recommend publication, and think the paper would be further improved by one more minor set of revisions.

Response: *We much appreciated your comments and suggestions. They really improved the quality of our work.*

R#4: ON THE INFLUENCE OF SEA-LEVEL CHANGE

R#4: While their geological framework is much improved, and their interpretations have sharpened significantly, I think they still downplay the effects of river long profile response to sea-level change. For me, this is the fundamental physical reality that provides context for the many valuable observations they make. And in the for what its worth category, they cite the Blum and Tornqvist (2000) paper on this issue, but I suggest the authors might find the Blum et al., 2013 Earth Science Reviews paper to be more specific about such issues.

Response: *Indeed, Blum et al. (2013) is a great review paper and fits much more with issues about valley incision and filling. Reference was updated.*

R#4: Regardless, both papers make the case that rivers have a predictable response to sea-level lowering - they incise their channels and valleys, as they extend their channels to lower sea-level and shoreline positions. This response tapers to zero upstream over a distance that scales to gradient and flow depth. Conversely, rivers have a predictable set of responses to sea-level rise - they must (a) aggrade and fill their valleys to keep pace with sea-level rise, or (b) form a drowned valley estuary if sediment supply is low. The distance over which this aggradation occurs also scales to gradient and flow depth - the minimum distance is roughly approximated by the backwater length which scales as $L_b \sim \text{mean water depth} / \text{slope}$. But, while the base of the channel responds upstream over that distance, the flood plain must also feel the effects. For example, if channel X is 25 m deep when the base of the channel descends below zero elevation, i.e. the backwater length, the flood plain height would be ~25 m elevation.

I agree with the authors that their Solimoes sites are pretty far upstream, but I would hypothesize that the river flood plain is still feeling the effects of base level rise. Coari, one of the places identified in the paper, is ~1850 river km from the mouth – I am not sure of the channel depth at that point, but the water surface in GoogleEarth is ~20 m elevation, so it is still likely within the reach that is affected by backwater hydraulics, and I would assume the aggradational drive from sea-level rise is still being felt there. Additionally, consider the gradient of the Solimoes is <0.00005 from Coari to Tefe. This is approximately the same

gradient the Amazon River downstream from Manuas must attain at sea-level lowstand to reach the shelf edge canyon at depths of ~100 m or so. My basic point is that it is plausible that flood plains throughout their entire study area feel the effects of climate-forced sea-level change.

Response: *Yes, we agree with this view that sea level effect can propagate upstream until western Amazon (Tefé) due to relatively low elevation and slope along the Solimões-Amazon pathway, and also until the lower reach of some lowland tributaries. The discussion topic was changed to reinforce this possibility, including the addition of a specific sentence (lines 270-277). Late Pleistocene incision followed by aggradation during the Holocene was observed in higher elevation terrains (> 100 m), such as in the middle Madeira, suggesting that fluvial hydraulics (backwater effect between tributary and trunk rivers) is also important for sediment accumulation in river valleys. The general idea is that incision and aggradation are driven by a lower frequency sea level (~100 ka) control, with decreasing intensity upstream, overlapped by the high frequency precipitation variation (<23 ka, precession) control. We hope that this view is now clarified in the discussion.*

R#4: For me, the incision and aggradation, which created the valley that has since been mostly filled with wetland environments like those discussed in the paper, is also part of the globally- and regionally-relevant climate story. I would argue this is the first-order explanation for the temporal distribution of their geochron data, including the minimal numbers of 14C or OSL ages during isotope stage 4, 3 and 2, and the peak number of ages during the late Holocene, as shown in Figures 6 and 7. But I would argue that such millennial-scale expansion of wetlands likely occurred during isotope stages 4,3 and 2, its just that the record is now buried. They mention this bias, but the methods they use are fundamentally biased to document deposits that represent global sea-level highstand. This does not diminish the significance of what they document in any way, what they have done stands on its own merit. But what they have documented could also be of a fundamental response of the river system's flood plain that likely occurred during all stages of sea level as the channel and flood plain adjust their profile to the contemporaneous sea-level and shoreline position.

Response: *The uppermost deposits of upland terraces (abandoned floodplains) are 20-30 m higher than the high water stage and have ages from 111-72 ka (lower deposits) to 63-45 ka (Pupim et al., 2019) (Fig. 6C). This indicates an aggradation phase and expanded floodplains under falling sea level (111-45 ka). Narrower floodplains with aggradation from a lower elevation level compared to paleo-floodplains point to an incision phase after 45 ka. Constraining the beginning of this new aggradation phase would depend on the access of basal deposits of the current floodplains. We agree that our geochronological dataset represents the minimum age of the beginning of this last aggradation phase, but terraces stratigraphy points to base level falling and reduction of floodplains between 45 ka and a period after the LGM. The variation in floodplain area is the critical landscape change for the discussed biotic and climate (methane) issues, independently of its major control (precipitation vs sea level). Here, ages from the archipelagos in the lower reach of flooded tributaries (Xingu, Tapajós and Negro) are the best data we got to constrain the beginning (or at least the acceleration) of this last aggradation phase because sampling is representative of the whole archipelago in two distant rivers (Xingu and Negro). However, aggradation should have a different pace in rivers with higher sediment supply, such as the Solimões and Madeira. In general, we agree that aggradation started before the Holocene*

under a lower base level, with buried records, but accelerated during the Holocene due to increasing sediment supply fueled by strengthening of the monsoon.

R#4: I also have long thought it likely that the many flooded river valleys that join the main channels are themselves drowned valleys because of aggradation of the main river channels – I do not know the literature enough to know whether this is incorrect or common knowledge. But I suspect they cannot keep up with the Andean sediment loads present in the main channels, so do not fill their valleys as fast as the Amazon has, but their very presence is likely also a consequence of climate-forced sea-level rise and drowning of glacial-period valleys.

Response: *Yes, we agree that sediment accumulation in Andean trunk rivers (Solimões and Madeira, for example) can lead to impoundment of tributaries with lower sediment loads. In parallel, there is also the backwater effect of the trunk river into the tributary, when water level is rising in the trunk river leading to flooding and hydraulic conditions favoring sediment accumulation in the tributaries. This is another problem requiring more specific stratigraphic, topographic/bathymetric and geochronologic data to be solved.*

R#4: In summary, I think changes in climate of the type discussed in the paper are very important, interesting, and solidly interpreted – what they discuss merits publication in Nature COMMS. But I also think all of those effects are overlain on this broader contextual stage that represents climate-forced global sea-level change and its effects on the Amazon river system, and it would be nice to see that featured in the paper as well.

Response: *We much appreciated your thoughtful comments. They really improved the arguments and clarified limitations. The text was revised in order to have more balance between the precipitation and sea level roles, including the title. Precipitation will accelerate the expansion of floodplains only if sea level is rising to increase accommodation space in the Solimões-Amazon valley, which is critical to make the flooded forests connected or disconnected across lowland Amazonia. So, sea level is the long term control while precipitation is a higher frequency control (orbital or millennial abrupt).*

R#4: SPECIFIC COMMENTS

R#4: I think the graphics are greatly improved, and, within a difficult environment, now display a lot of the important context for the geochron data that underlies their interpretations.

Response: *Thanks.*

R#4: I think they need a general location map for the main text. As it is now, the reader has to go to the Supplementary Material, Figure S1, to see the locations of study areas within the context of the river system. Figure 1 in the text just shows local context, which is very useful, but the reader will have to go to the Figure S1 to see where the locations are.

Response: *We improved the general map and moved it from supplementary material to the main text. It is now Figure 1.*

Reviewed by Mike Blum